# SPLiT: Popularity-Bias-Aware Online Prompt Optimization for LLM-based Recommendation

## Abstract

Large Language Model (LLM)-based recommender systems often rely on preference summaries to condense a user's interaction history and help the model better capture the user's interests. The quality of downstream recommendations depends heavily on how accurately the preference summaries align with true preferences. However, prior work has overlooked popularity bias in these summaries, which often over-represents popular items, and thus, recommendation quality degrades. Moreover, the inherent randomness of LLMs produces summaries with varying fidelity and bias. To address this, we propose an online learning approach that identifies the most accurate and least biased preference summary. We formulate the preference summary selection task as a Contextual Bayesian Optimization with Constrained Set problem and introduce the Semantic Popularity Lift-based Preference Summary selecTion (SPLiT) framework. SPLiT incorporates a Semantic Popularity Lift penalty that quantifies how much a summary amplifies popularity bias. The penalty discourages selecting high-bias summaries and guides the choice toward those that better reflect the user's true preferences. SPLiT significantly improves recommendation performance by mitigating popularity bias, achieving 13.8% higher Normalized Discounted Cumulative Gain and 6.9% higher Hit Rate compared with the best baseline. This highlights the importance of popularity bias-aware summary selection for debiasing prompt optimization, advancing fairness and accuracy in LLM-based recommender systems.

## 1 Introduction

Recommender systems play a key role in platforms, e.g., video streaming, e-commerce, and social media, by helping users discover relevant content. Conventional approaches, however, often rely on collaborative filtering or frequency-based methods (He et al., 2017), which exploit user-item interaction history patterns to generate recommendations. However, they face fundamental challenges such as the cold-start problem, since these methods rely largely on observed user–item interactions and struggle when such data are sparse (Lam et al., 2008; Li et al., 2010). To address these challenges, LLM-based recommender systems have emerged. Unlike traditional approaches, LLMs have broad world knowledge and natural-language reasoning that can capture potential relevance beyond observed interactions. This capability enables preference inference under sparse user–item interactions and improves recommendation quality (Dai et al., 2023; Hou et al., 2024). Building on these capabilities, LLM-based recommender systems have recently emerged as a promising approach for accurate and personalized recommendations (Liu et al., 2023a; Dai et al., 2023; Gao et al., 2025).

As shown in Figure 1, typical LLM-based recommender systems construct prompts that incorporate user profiles, interaction history, item metadata (e.g., genres or tags, where items refer to the entities being recommended, such as movies in movie recommender systems), and other contextual information. These prompts are then fed into the recommender LLM to generate recommendation results (Dai et al., 2023). By integrating world knowledge with user context and item metadata, and leveraging natural-language reasoning, LLMs can effectively utilize diverse information sources to achieve higher recommendation quality than traditional methods (Liu et al., 2023c).

However, LLM-based recommender systems suffer from issues such as popularity bias, which tends to over-recommend generally popular items or genres (Deldjoo, 2024), resulting in skewed recommendations and thus degraded recommendation performance (Lichtenberg et al., 2024). This

popularity bias arises from the tendency of LLMs to memorize their training datasets Di Palma et al. (2025). When popular items dominate the data, this memorization skews recommendations toward those popular items, underscoring the need for effective bias mitigation strategies.

To address this, prior work has proposed different approaches to improve recommendation performance for LLM-based recommender systems (Hua et al., 2023). One effective approach is to introduce debiasing instructions; this approach adds specific guidance into the input prompt template of the recommender LLM to alleviate popularity bias (Lichtenberg et al., 2024). Another approach is to integrate a preference summary. Prior work shows that a natural-language summary of user context (e.g., interaction history), which we refer to as a preference summary, can be generated through a summarization LLM. This preference summary provides a condensed representation of user preferences. Incorporating the summary into the input prompt of the downstream recommender enables the recommender LLM to better capture user preferences (Kusano et al., 2024). Moreover, how accurately the preference summary aligns with the true user context directly influences the performance of downstream recommendations (Wang et al., 2025). While this preference summary operates orthogonally to debiasing instructions, both approaches share the same objective of improving recommendation performance, and recent work further shows that combining preference summaries with debiasing instructions can achieve additional gains (Hamad, 2025).

Although popularity bias can be alleviated in the recommender LLMs, we observe that the inherent bias of LLMs can also affect the summarization process itself (see Section 4.2). When generating a preference summary using a summarization LLM, the preference summary may over-represent popular items or genres, thereby introducing popularity bias in the preference summary as well. This biased summarization propagates into the recommender LLM, ultimately degrading recommendation performance. Crucially, existing works have overlooked this problem: while they focus on alleviating popularity bias in the recommender LLM, they neglect the bias already embedded in the preference summary produced by the summarization LLM. This reveals a key challenge: preference summaries often over-represent popular items or genres, making it difficult to capture user-specific preferences. This raises our main goal: *how to mitigate the popularity bias in preference summary so that it more stably captures true preferences of users and improves recommendation performance?*

Prior work commonly generates a preference summary by using LLMs to summarize the user context (e.g., interaction history) (Kusano et al., 2024). However, we observe that the inherent randomness of LLMs leads to different preference summaries across runs (see Appendix E.2. Specifically, these summaries vary both in how accurately they capture user preferences and in the extent to which they amplify popularity bias. Motivated by this, we aim to identify and select higher-quality preference summaries that better align with user context while exhibiting lower bias, thereby mitigating the popularity bias introduced during LLM-based generation and improving recommendation performance. Compared with traditional prompt optimization methods, such as black-box optimization using LLMs (Cheng et al., 2023; Madaan et al., 2023), the selection approach takes a different direction. It optimizes preference summary and mitigates its popularity bias in a more reliable and numerically interpretable manner (Shi et al., 2024; Ramnath et al., 2025; Di Palma et al., 2025).

In this paper, we formulate the preference summary selection as a new optimization problem, Contextual Bayesian Optimization with a Constrained Set (CBO-CS). Building on this formulation, we propose SPLiT (**S**emantic **P**opularity **Li**ft-based Preference Summary selec**T**ion), a novel approach that addresses this problem and mitigates popularity bias in LLM-based recommender systems. The key idea is to optimize the preference summary through bias-aware selection. Beyond maximizing reward (user feedback such as click-through rate or ratings, instantiated in our experiments with metrics like Normalized Discounted Cumulative Gain and Hit Rate), we introduce Semantic Popularity Lift (SPL) to measure how much a candidate preference summary amplifies popularity bias relative to the user's history. By balancing predicted reward with the SPL, SPLiT selects summaries that both align with user context and reduce the tendency to over-recommend popular items. Unlike prior works that address popularity bias by prompt tuning for the recommender LLM (Lichtenberg et al., 2024; Gao et al., 2025), we focus on mitigating popularity bias within the preference summary, ensuring that the summary is less skewed before being fed into the recommender LLM. This design introduces a new perspective on mitigating popularity bias to quantify the bias and guide the selection of preference summaries, thereby providing a more interpretable debiasing mechanism in LLM-based recommendations. Our main contributions can be summarized as follows:

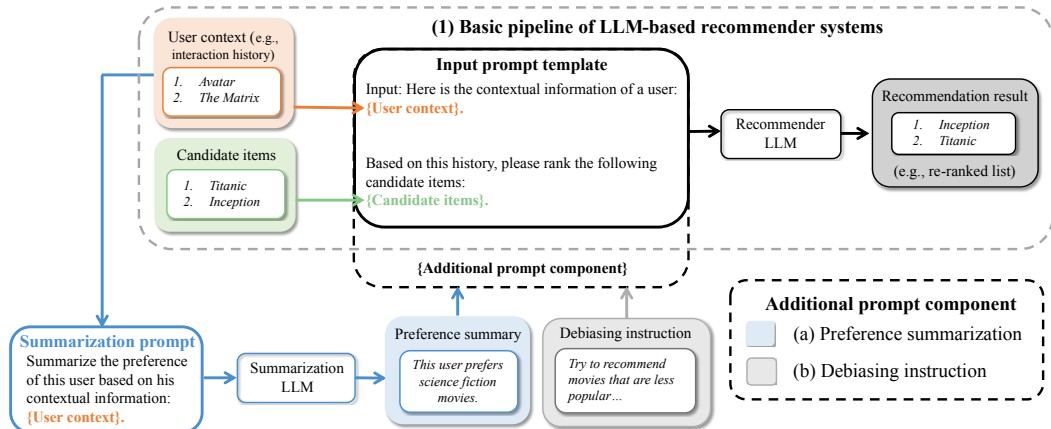

Figure 1: Baseline pipelines of LLM-based recommender systems. (1) Basic pipeline: the user context (e.g., interaction history) and candidate items are directly fed into the recommender LLM to produce recommendation results (Dai et al., 2023). (2) Optimized pipeline: the prompt is augmented with an additional component. (a) Preference summarization: a summarization LLM generates a natural-language summary from the user context, which is appended to the prompt (Kusano et al., 2024). (b) Popularity-bias mitigation: a debiasing instruction is added to the input prompt template to reduce popularity bias (Lichtenberg et al., 2024).

(i) We reveal that LLM-generated preference summaries tend to over-represent popular items and genres, which degrades downstream recommendation quality and amplifies popularity bias. To analyze this overlooked source of bias, we introduce the new SPL metric, which quantifies whether preference summaries over-represent popular genres compared with user histories. Analysis with SPL shows that preference summaries exhibit such bias, highlighting the need to address it when improving recommendation performance and fairness.

(ii) We formulate a new problem, CBO-CS, for optimizing preference summaries through selection to better align with user context, and develop SPLiT to solve it in LLM-based recommender systems. SPLiT addresses popularity bias by introducing an SPL-based penalty into the selection process, which discourages preference summaries that over-represent popular genres and guides the selection away from summaries with popularity bias.

(iii) We evaluate SPLiT in real-world recommendation scenarios using the MovieLens-1M and Last.fm dataset. The results show notable improvements over multiple baselines. First, compared with other selection methods, SPLiT achieves the lowest cumulative regret and the smallest SPL value, demonstrating higher accuracy in selecting the optimal preference summary while reducing popularity bias. Second, when applied to LLM-based recommender systems, SPLiT improves NDCG by 13.8% and HR by 6.9% over the best baseline, which indicates substantial gains in recommendation performance.

## 2 RELATED WORK

In this section, we focus on recent work that solves the prompt selection and the popularity bias problems. We further introduce the existing work on recommender systems and prompt optimization methods in Appendix A.

**Prompt Selection.** As an interpretable approach to prompt optimization (Chen et al., 2023), prompt selection makes the decision process transparent by explicitly evaluating candidate prompts, rather than relying on traditional black-box optimization through LLMs (Lin et al., 2024). Early methods typically assumed a fixed set of candidate prompts for selection (Mao et al., 2025; Wu et al., 2024b), overlooking the process of generating prompts from contextual information. Recent studies address this limitation by formulating prompt selection as an end-to-end optimization process, often modeled as a Bayesian Optimization problem (Schneider et al., 2024; Shi et al., 2024). Despite these

advancements, most existing works treat the prompt generation and selection stages as independent problems (Do et al., 2025; Lin et al., 2024). This separation overlooks the strong interdependence between the two stages and their mutual influence on overall performance. In particular, when LLMs are used as prompt generators, they often exhibit inherent limitations such as popularity bias (Lichtenberg et al., 2024; Spurlock et al., 2024; Zhang et al., 2024), which can degrade the quality of the chosen prompt. These observations highlight the need for unified approaches that jointly model prompt generation and selection while accounting for such inherent limitations.

**Popularity Bias.** Popularity bias has been studied as a critical fairness issue in recommender systems (Klimashevskaia et al., 2024). In LLM-based recommender systems, recent studies have shown that inherent limitations of LLMs make them tend to over-recommend popular items (Lichtenberg et al., 2024; Sakib & Das, 2024), primarily due to memorization effects in LLMs (Di Palma et al., 2025). When the training data is imbalanced across items, such memorization can lead to recommendations skewed toward high-frequency items. A common metric for quantifying this bias is popularity lift, which measures the gap between the popularity distribution of recommended items and that of actual user interaction history (Abdollahpouri et al., 2020). Given the significance of this issue, recent research has explored approaches such as prompt tuning to mitigate popularity bias (Lichtenberg et al., 2024; Bito et al., 2025). However, prior work on popularity bias in LLM-based recommenders has focused on recommendation results, leaving the bias embedded in preference summaries unexplored. Our work aims to identify, analyze, and mitigate this overlooked source of popularity bias, thereby filling an important gap in the literature.

## 3 System Model

In this section, we present the overall architecture of an LLM-based recommender system with preference summary selection (see Figure 2). Following Kusano et al. (2024) and Guo et al. (2024), how accurately a preference summary captures the user's context strongly influences recommendations. This motivates us to focus on selecting the candidate preference summary most aligned with the user's true preferences to improve recommendation performances. Additional analysis supporting our design choice of selecting among candidate preference summaries rather than final recommendation lists is provided in Appendix E.7. In our recommender system setting, the process unfolds over a finite time horizon of $T$ rounds. Each round $t \in \{1, 2, \cdots, T\}$ corresponds to the recommendation process for one user. At each round $t$, the following sequence of events occurs.

**Context and Preference Summary Generation.** The system first observes a user context $c_t$ (e.g., a user's interaction history). It then uses a candidate summary generator $G$, which takes $c_t$ as input, to produce $n$ preference summaries $p_{t,1}, p_{t,2}, \ldots, p_{t,n}$, where $n$ is a fixed hyperparameter. These summaries are natural-language descriptions of user preferences derived from the context. Together, they form the candidate summary set $\mathcal{A}_t(c_t) = \{p_{t,1}, p_{t,2}, \cdots, p_{t,n}\}$. In practice, $G$ is typically implemented as an LLM-based generator (Do et al., 2024), with implementation details for our setting provided in Appendix B.1.

**Result and Reward Generation.** After generating the candidate summary set $\mathcal{A}_t(c_t)$, the system selects one preference summary $p_t \in \mathcal{A}_t(c_t)$ for the current round. The selected summary, together with the context $c_t$, is fed into a downstream recommender LLM, which we denote by a stochastic function $Q$, to output a recommendation result $q_t \sim Q(p_t, c_t)$. A scalar reward $r_t$ is then computed by a function $r(q_t, c_t)$ that takes the recommendation result and the user context as input and outputs reward. Importantly, only the reward of the chosen summary is observed, while the rewards of unchosen candidates remain unknown.

While prior research on popularity bias in LLM-based recommender systems has mainly focused on the recommender LLM, we highlight that the LLM-based candidate summary generator $G$ can also introduce popularity bias into the generated preference summaries. This overlooked source of bias motivates the need to address popularity bias in the summarization phase.

## 4 SPL: Metric and Problem Formulation

Building on the system model in Section 3, we now focus on the challenge of popularity bias in preference summaries. To quantify this effect, we first introduce the Semantic Popularity Lift (SPL)

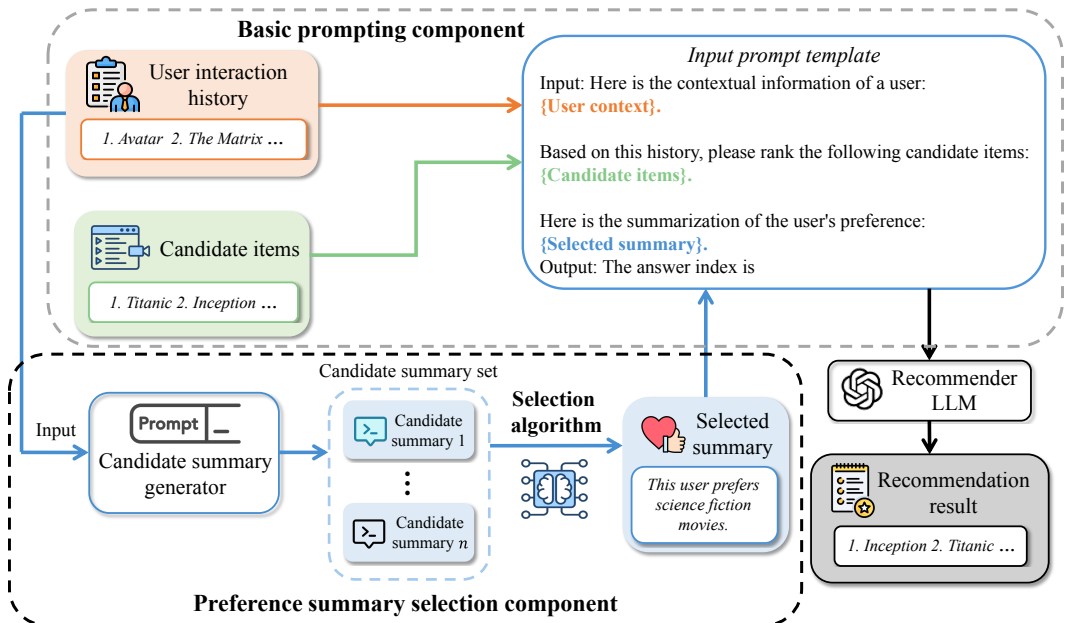

Figure 2: Pipeline of an LLM-based recommender system with preference summary selection. The figure contains two components: (i) the Basic prompting component, which incorporates an LLM-generated summary without selection, following the Kusano et al. (2024) pipeline; and (ii) the Preference summary selection component, which generates multiple candidate summaries and selects.

metric in Section 4.1, which measures the popularity bias of preference summaries. Leveraging this metric, we present key observations in Section 4.2, showing that preference summaries not only exhibit popularity bias but also vary in fidelity due to the inherent randomness of LLMs. Finally, in Section 4.3, we formulate the preference summary selection task as a new optimization problem, Contextual Bayesian Optimization with Constrained Set (CBO-CS), and set the objective of learning an online policy that mitigates popularity bias in order to improve recommendation performance.

### 4.1 SPL: POPULARITY BIAS METRIC

In this section, we design a new metric, SPL, to quantify the popularity bias embedded in preference summaries. In prior work, the metric of popularity lift has been used to show the popularity bias of recommendation items by measuring the extent to which recommended items include disproportionately more popular genres compared to a user's interaction history (Abdollahpouri et al., 2020). However, this metric does not directly apply to preference summaries, since it is defined based on the frequency of genres in recommendation results. In contrast, a preference summary is a natural-language description rather than a list of items, and such frequencies cannot be directly quantified. To address this limitation, we propose SPL, which evaluates how much an LLM-generated preference summary semantically amplifies popularity bias relative to the user's history. To the best of our knowledge, SPL is the first metric designed for capturing popularity bias in preference summaries. To this end, we first introduce two expectations: $E_{pr}(p)$, the summary preference expectation inferred from the preference summary $p$, and $E_h(c)$, the user history preference expectation inferred from the user's context $c$.

**Global Genre Popularity.** Following Abdollahpouri et al. (2020), let $\Gamma$ be the set of genres and $\gamma \in \Gamma$ an index. We maintain global popularity counts $\{N_\gamma\}_{\gamma \in \Gamma}$, where $N_\gamma$ denotes the number of times genre $\gamma$ has appeared, and total $N_{\text{tot}} = \sum_{\gamma \in \Gamma} N_\gamma$. We define the global genre popularity as $\theta(\gamma) := N_\gamma / N_{\text{tot}}$, which is updated online from newly observed contexts.

**Semantic Textual Similarity Distributions.** For a round with context $c$ (a user's interaction history) and a preference summary $p$, We employ a Semantic Textual Similarity Model (STS), instantiated with the nli-roberta-base model (Reimers & Gurevych, 2019), denoted as $\text{STS}(\cdot, \cdot)$. The

STS model takes as input a text (either the context $c$ or the preference summary $p$) and a genre label $\gamma \in \Gamma$, encodes both into sentence embeddings, and computes their similarity (e.g., cosine similarity). A softmax normalization is then applied to obtain a probability distribution over genres:

$$w_h(c, \gamma) := \mathsf{STS}(c, \gamma), \quad w_{pr}(p, \gamma) := \mathsf{STS}(p, \gamma), \tag{1}$$

with $\sum_{\gamma \in \Gamma} w_h(c, \gamma) = \sum_{\gamma \in \Gamma} w_{pr}(p, \gamma) = 1$. Here $w_h$ denotes the genre preference distribution implied by the user's history, while $w_{pr}$ reflects the genre tendency of the preference summary.

**User History Preference Expectation.** Let $c$ denote the user context information; in our setting, we use the interaction history as the context. The user history preference expectation, denoted as $E_h(c)$, is defined as

$$E_h(c) := \sum_{\gamma \in \Gamma} w_h(c, \gamma)\, \theta(\gamma) = \mathbb{E}_{\gamma \sim w_h(c, \cdot)}\big[\theta(\gamma)\big]. \tag{2}$$

Here, $w_h(c, \gamma)$ is the normalized weight that the STS model assigns to genre $\gamma$ given the user's context $c$, and $\theta(\gamma)$ is the global popularity of genre $\gamma$. Thus, $E_h(c)$ is the average global popularity value of genres according to the user's historical preferences. Higher values of $E_h(c)$ mean their preferred genres are popular overall, while lower values indicate a tendency toward niche genres.

**Summary Preference Expectation.** Let $p$ denotes a preference summary. The summary preference expectation, denoted as $E_{pr}(p)$, is defined as

$$E_{pr}(p) := \sum_{\gamma \in \Gamma} w_{pr}(p, \gamma)\, \theta(\gamma) = \mathbb{E}_{\gamma \sim w_{pr}(p, \cdot)}\big[\theta(\gamma)\big]. \tag{3}$$

Here, $w_{pr}(p, \gamma)$ is the normalized weight that the STS model assigns to genre $\gamma$ given the preference summary $p$, and $\theta(\gamma)$ is the global popularity of genre $\gamma$. Thus, $E_{pr}(p)$ is the average global popularity value of genres according to the preference summary's genre-preference distribution. Higher values of $E_{pr}(p)$ mean the preference summary biases toward popular genres, while lower values indicate a tendency toward niche genres.

**Semantic Popularity Lift (SPL).** We define the SPL to quantify the popularity bias embedded in preference summaries. Formally, SPL compares the relative difference between these two expectations:

$$\mathrm{SPL}(p, c) = \frac{E_{pr}(p) - E_h(c)}{E_h(c)}. \tag{4}$$

If $\mathrm{SPL}(p, c) > 0$, the summary amplifies popularity bias; if $\mathrm{SPL}(p, c) < 0$, it emphasizes popular items less than the user history; and if $\mathrm{SPL}(p, c) = 0$, no amplification occurs. Larger absolute values indicate greater deviation in popularity bias relative to the history. As further discussed in Appendix C.1, the SPL measured on preference summaries is consistent with the traditional popularity lift measured on recommendation results. Moreover, based on the analysis in Appendix C.2, we observe that a large SPL value indicates stronger popularity bias in the preference summary, which in turn leads to degraded recommendation performance. These correspondences validate SPL as an appropriate metric for capturing the popularity bias of preference summaries.

## 4.2 Key Observations and Motivation

We investigate the behavior of LLM-generated preference summaries and derive key observations that motivate our problem formulation and algorithmic design. These observations reveal both the presence of popularity bias and the variability introduced by LLM randomness, providing the foundation for our selection-based approach to mitigate popularity bias of preference summaries.

**Observation 1 (Popularity bias in LLM-generated preference summary).**

The preference summary is designed to capture user history in a natural-language form. However, LLM-generated preference summaries may often over-represent popular items or genres, thereby introducing popularity bias. As illustrated in Figure 7 (see Appendix E.3), popular genres dominate the generated summaries even when they are underrepresented in the user's history. To evaluate this effect, we employ the SPL metric introduced in Section 4.1 to calculate the popularity bias of preference summaries across user groups. Figure 3 reports SPL values across user groups partitioned by their history preference expectation $E_h(c)$, which measures the average global popularity

of genres in a user's interaction history. Across all groups, SPL values remain consistently above zero, indicating that LLM-generated summaries tend to amplify popularity bias. The effect is particularly strong for users with niche tastes, which exhibit higher average SPL, while users with more mainstream preferences show relatively smaller bias amplification. These findings show that popularity bias is indeed present in LLM-generated preference summaries, underscoring the importance of addressing it to improve both the performance and fairness of LLM-based recommendations.

**Obeservation 2 (Variability induced by LLM randomness).** Beyond bias, another challenge arises from the inherent randomness of LLMs. Even under the same user context, different runs of the summarization LLM may produce preference summaries that vary widely in both fidelity to user preferences and the degree of popularity bias, causing different recommendation performances. Some generated summaries align closely with user history, while others strongly amplify popular genres. The examples of experimental results are shown in the Appendix E.2. These observations suggest that a selection-based approach can be used to mitigate the popularity bias of preference summaries and thereby improve recommendation performance.

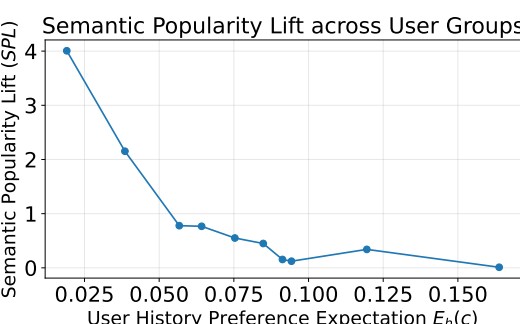

Figure 3: Popularity-bias amplification by preference summarization across user groups. SPL values above zero indicate amplification, with stronger effects for niche-taste groups (low user history preference expectation).

### 4.3 PROBLEM FORMULATION

We now describe our online preference summary selection problem as a novel CBO-CS problem. The goal is to improve recommendation performance by mitigating the popularity bias of preference summaries through selection. Following the system model in Section 3, the process unfolds over $T$ rounds, indexed by $t \in \{1, 2, \cdots, T\}$. In each round $t$, the system observes a user context $c_t$ and a set of candidate summaries $\mathcal{A}_t$. An online policy $\pi$ chooses one summary $p_t \in \mathcal{A}_t$ based on $c_t$ and past observations. Executing summary $p_t$ yields a recommendation from the downstream recommender LLM and an observed reward $r_t$. Let $\mu(p, c)$ denote the (unknown) expected reward when applying summary $p$ under context $c$. For each round, the optimal achievable mean reward is $\mu_t^\star = \max_{p \in \mathcal{A}_t} \mu(p, c_t)$.

**Regret.** The performance of a policy is evaluated using cumulative regret over $T$ rounds, which measures the gap between the policy and an oracle that always selects the best summary:

$$\text{Regret}(T) = \mathbb{E}\left[\sum_{t=1}^{T} \left(\mu_t^\star - r_t\right)\right],$$

**Objective.** The goal is to find an online policy $\pi$ from the space of admissible policies $\Pi$ that minimizes regret:

$$\min_{\pi \in \Pi} \text{Regret}(T),$$

Intuitively, this objective ensures that the system quickly identifies the preference summary that aligns with user interests, so that the overall recommendations approach the quality of those generated by an oracle with perfect knowledge.

## 5 ALGORITHM DESIGN

In this section, inspired by Observation 1 in Section 4.2, which shows that preference summaries exhibit popularity bias, we build on the SPL metric introduced in Section 4.1 to develop our algorithm. We then leverage this metric in introducing Semantic Popularity Lift-based Preference Summary Selection (SPLiT), which serves as an algorithmic solution to the CBO-CS formulation presented

in Section 4.3. SPLiT addresses the constrained, context-dependent preference summary selection problem while simultaneously mitigating the popularity bias induced by candidate preference summaries. Building on Section 4.1, our key idea is to select the preference summary that maximizes the expected reward while penalizing those that over-represent popular genres, i.e., the preference summary with high SPL. The pseudocode of SPLiT is in Algorithm 1 (see Appendix B.5).

At each round $t$ with context $c_t$ (the user's interaction history at round $t$), the algorithm proceeds as follows. The preference summary generator $G$ first produces a candidate set $\mathcal{A}_t(c_t)$ of size $n$ (line 3 of Algorithm 1). For every candidate preference summary $p \in \mathcal{A}_t(c_t)$, the reward estimator $\mathcal{M}$ predicts the mean reward $\hat{\mu}_t(p)$ (line 5 of Algorithm 1). The SPL penalty $\xi_t(p)$ is then computed by Algorithm 2 (line 6 of Algorithm 1), which compares the preference summary preference expectation $E_{pr}(p)$ with the user history preference expectation $E_h(c_t)$ as defined in Section 4.1. Given these components, SPLiT assigns each preference summary a penalized score $s_t(p) = \hat{\mu}_t(p) - \lambda\xi_t(p)$ (line 7 of Algorithm 1), where $\lambda$ is a tunable trade-off hyperparameter balancing the predicted reward and the SPL penalty. The algorithm then selects the preference summary $p_t$ that maximizes this score (line 8 of Algorithm 1). Through this step, we are able to optimize for summaries with higher expected reward while simultaneously favoring those with lower popularity bias. The chosen preference summary $p_t$ is then passed to the downstream recommender $Q$ to produce a recommendation $q_t$ (line 9 of Algorithm 1). After receiving $q_t$, the user reacts through an interaction such as clicking, watching, or rating, which is converted into a scalar reward signal $r_t = r(q_t, c_t)$ by the unknown reward function $r$ (line 9 of Algorithm 1). This reward $r_t$ serves as the explicit feedback for online updates. The reward estimator $\mathcal{M}$ is subsequently updated using the observed context $c_t$, reward $r_t$, selected preference summary $p_t$, and the updated history $\mathcal{H}_t$ (lines 10-11 of Algorithm 1), where $\mathcal{H}_t$ stores all past observations up to round $t$. The global popularity counts $N_\gamma, N_{\text{tot}}$ are updated using the observed context $c_t$ by Algorithm 3 (lines 12 of Algorithm 1). Together, Algorithms 2 and 3 are invoked inside Algorithm 1 (line 6 and the final line of Algorithm 1, respectively), ensuring that SPL penalty and global popularity updates are integrated into the SPLiT algorithm.

# 6 EXPERIMENTS

In this section, we will evaluate our proposed SPLiT for LLM-based recommender systems and compare it with other baselines. We assess performance along two aspects. (i) Optimization effectiveness in Section 6.1: we compare SPLiT against standard Bayesian-optimization baselines and report cumulative regret under the CBO-CS problem. (ii) Recommendation quality in Section 6.2: we compare SPLiT with LLM-based prompt-optimization methods, including approaches designed to mitigate popularity bias in Lichtenberg et al. (2024), using standard ranking metrics.

**Datasets and Tasks.** We conduct experiments on two widely used datasets: MovieLens-1M (Harper & Konstan, 2015) and Last.fm (Bertin-Mahieux et al., 2011). We adopt the standard Top-$K$ recommendation task (Appendix D.1) and follow a leave-one-out evaluation strategy (Dai et al., 2023) (Appendix D.3). For each user, the most recent interaction is used as the test item, while the 2–6 preceding interactions form the interaction history. To construct the candidate set, we apply BM25 (Robertson et al., 1995) (Appendix B.3) to retrieve four candidate items from the corresponding dataset (movies in MovieLens and tracks in Last.fm), which are then combined with the held-out test item to form a five-item candidate set.

**Model Specification.** The recommender LLM is implemented using GPT-4o-mini (OpenAI et al., 2024), and generates a ranked list over the candidate set as the recommendation results for the target user. For all experiments, we repeat the evaluation five times and report average results, where each run randomly samples 100 users from the MovieLens-1M dataset. A recommendation to one user corresponds to a single decision round.

**Evaluation Metrics.** We evaluate our framework with two types of metrics: (i) *Selection metric*: To assess the effectiveness of preference summary selection, we report cumulative regret (Eq. (**??**)), which measures the expected reward loss of the selection policy compared with an oracle policy that always chooses the optimal summary at each round. This metric focuses on how efficiently the selection algorithm identifies an optimal preference summary. (ii) *Recommendation metrics*: To assess recommendation quality, we use two standard metrics following common practice: NDCG@$k$ (Normalized Discounted Cumulative Gain; (Järvelin & Kekäläinen, 2002)) and HR@$k$ (Hit Rate),

Table 1: Final cumulative regret and Semantic Popularity Lift (SPL) over 100 rounds.

| Metric | GP-UCB | GP-EI | BNN-UCB | BNN-EI | SPLiT (ours) |
|---|---|---|---|---|---|
| Cumulative Regret | 4.521 | 4.569 | 4.326 | 4.505 | **3.726** |
| SPL | 0.455 | 0.462 | 0.456 | 0.453 | **0.438** |

which measures the proportion of users for whom the test item appears in the top-$k$ positions of the ranked list. Detailed definitions are provided in Appendix D.4 and D.5.

## 6.1 OPTIMIZATION EFFECTIVENESS

In this section, we evaluate the effectiveness of SPLiT as a preference summary selection algorithm for solving the CBO-CS problem in Section 4.3. To assess its performance, we compare SPLiT against baseline methods designed for Bayesian Optimization problems. We adopt the NDCG@3 as the reward defined in Section 3, and we will calculate the cumulative regret for each algorithm.

**Baselines.** We compare SPLiT against several standard Bayesian optimization baselines: (i) GP-UCB (Rasmussen, 2003; Srinivas et al., 2009), (ii) GP-EI (Rasmussen, 2003; Jones et al., 1998), (iii) BNN-UCB (Springenberg et al., 2016), and (iv) BNN-EI (Springenberg et al., 2016). Further details of the optimization baselines are provided in Appendix D.6.

**Experimental results.** We conduct experiments on the MovieLens-1M dataset and evaluate optimization performance using cumulative regret as the primary metric. A lower cumulative regret indicates that the algorithm can more effectively identify a high-quality preference summary with fewer suboptimal selections. Figure 8 shows the regret curves of our method and the baselines, where our method achieves lower regret across rounds. We further summarize the final regret values after 100 rounds in Table 1: SPLiT obtains the lowest final regret (3.726), outperforming GP-based methods (4.521 for GP-UCB, 4.569 for GP-EI) and BNN-based methods (4.326 for BNN-UCB, 4.505 for BNN-EI). These results imply that our SPLiT converges more quickly to select an optimal preference summary, requiring fewer suboptimal selections to approach the oracle performance.

Beyond cumulative regret, we also evaluate algorithms using the SPL metric introduced in Section 4.1, which quantifies how strongly the preference summary amplifies popularity bias. Table 1 shows that SPLiT again achieves the lowest SPL value (0.438), compared with 0.455–0.462 for the baselines. This indicates that SPLiT more effectively mitigates the popularity bias while preserving strong optimization performance. Taken together, these results highlight that SPLiT not only minimizes regret to guide preference summary optimization toward better-performing candidates, but also provides fairer, less bias-skewed recommendations than existing methods.

## 6.2 RECOMMENDATION QUALITY

We further evaluate recommendation quality against representative LLM-based recommender systems (Dai et al., 2023) and methods designed to mitigate popularity bias (Lichtenberg et al., 2024).

**Baselines.** We evaluate SPLiT against representative LLM-based recommenders and recent popularity-bias-mitigation methods: (i) LLM4RS (Dai et al., 2023), (ii) LLMRank (Hou et al., 2024), (iii) LLM4Rerank (Gao et al., 2025), and (iv) WOK (Lichtenberg et al., 2024). A detailed description of all baseline methods is provided in Appendix D.7.

**Experimental Results.** We conduct experiments on the MovieLens-1M dataset and Last.fm dataset. Following the experimental setup described in Section 6, we evaluate the proposed SPLiT algorithm together with the baseline methods in the context of LLM-based recommender systems. The evaluation is conducted using four standard ranking metrics: NDCG@1/3/5, and HR@3.

In Table 2 and 6, we report a comparison of SPLiT against the baselines on the MovieLens-1M and Last.fm datasets respectively. The results show that SPLiT consistently outperforms all baselines. In particular, on the MovieLens-1M dataset, SPLiT achieves an improvement on NDCG@1 (+13.82%) and HR@3 (+6.85%), indicating its ability to place the most relevant items at the very top of the ranking and increase the likelihood of recommending items that users actually engage with. The

Table 2: Performance comparison of SPLiT and baseline methods on the MovieLens-1M dataset with GPT-4o-mini model. We report NDCG@1, NDCG@3, NDCG@5, and HR@3.

| Method | NDCG@1 | NDCG@3 | NDCG@5 | HR@3 |
|---|---|---|---|---|
| LLM4RS | 0.2812 | 0.5171 | 0.6476 | 0.6875 |
| LLMRank | 0.2667 | 0.5008 | 0.6318 | 0.6800 |
| LLM4Rerank-accuracy | 0.3540 | 0.5446 | 0.6739 | 0.6880 |
| LLM4Rerank-fairness | 0.3300 | 0.5475 | 0.6658 | 0.7100 |
| WOK-minimization | 0.2433 | 0.4755 | 0.6188 | 0.6633 |
| WOK-mitigate | 0.3567 | 0.5612 | 0.6886 | 0.7367 |
| **SPLiT (ours)** | **0.4060** | **0.6175** | **0.7101** | **0.7806** |
| *Improvement over best baseline* | *+13.82%* | *+10.03%* | *+3.12%* | *+6.85%* |

gains on NDCG@3/5 further demonstrate that the advantage of SPLiT extends beyond the very top ranks, yielding more accurate rankings overall. These results demonstrate that SPLiT better captures user preferences and delivers higher-quality recommendations by mitigating popularity bias.

# 7 DISCUSSION

Our study highlights that popularity bias does not only manifest at the level of recommendation outputs, but also at the level of LLM-generated prompts, which fundamentally shape downstream recommendations. By explicitly quantifying this bias through the SPL, our proposed SPLiT algorithm provides a transparent and interpretable mechanism for selecting prompts that better align with users' true preferences. This marks a shift from traditional prompt tuning or heuristic-based approaches, which often operate as black boxes and lack explicit control over bias.

Nevertheless, several challenges remain. First, while SPLiT demonstrates effectiveness in mitigating popularity bias, its reliance on semantic similarity models such as STS introduces dependencies that may affect robustness across domains. Second, our experiments are conducted on two benchmark datasets; extending the evaluation to larger-scale, real-world environments would further validate the generalizability of the approach. Third, although SPL captures popularity bias at the prompt level, other biases (e.g., exposure bias and position bias) may interact with it in complex ways, which our framework does not address yet. Moreover, although the candidate preference summary generator used in Appendix B.1, which follows a self-refine strategy (Madaan et al., 2023), is effective in improving recommendation performance, it can be costly and slow. Finally, while our experiments focus on the movie and music recommendation scenario, the issue of popularity-skewed preference summarization is general and can arise in other recommendation tasks such as news or e-commerce, suggesting that SPLiT may be broadly applicable.

# 8 CONCLUSION

We investigated preference summary selection for LLM-based recommender systems, focusing on the central challenge of popularity bias, which distorts fairness and limits the capture of user preferences. To address this, we formulate the task as the CBO-CS problem and propose SPLiT . By introducing the SPL penalty, SPLiT offers a transparent mechanism to quantify how candidate prompts amplify popularity bias, discouraging over-recommendation of popular genres while better aligning with niche user preferences. Experiments show that SPLiT reduces cumulative regret and mitigates bias, ultimately leading to improved recommendation performance. These results highlight bias-aware preference summary selection as a promising direction for LLM-based recommendation.

In future work, we will extend SPLiT to incorporate multiple forms of bias simultaneously, creating a unified framework for bias-aware prompt optimization. Furthermore, it would be interesting to integrate user feedback more dynamically, allowing the system to adaptively refine the trade-off between reward and bias mitigation in response to changes in user preferences. Finally, investigating cross-domain and multimodal recommendation scenarios may reveal broader applicability, especially in settings where textual prompts interact with images, audio, or structured metadata.

ETHICS STATEMENT

This work does not involve human subjects, personally identifiable information, or sensitive attributes. All datasets used are publicly available and widely adopted in prior research. We are aware that large language models may amplify popularity bias in recommendation tasks. To mitigate such risks, we explicitly analyze bias in our experiments.

REPRODUCIBILITY STATEMENT

We take reproducibility seriously and provide extensive details to facilitate replication: (i) dataset descriptions are given in Appendix D.2; (ii) full hyperparameter settings and training details are listed in Appendix B.4; (iii) pseudocode of the core algorithm is included in Appendix B.5; and An anonymous repository with code and scripts will be released with the camera-ready version.

LLM USAGE

Large Language Models (LLMs) were integral to the methodology of this paper. Specifically, we used GPT-4o-mini as the summarization model to generate candidate preference summaries and as the downstream recommender model to produce ranked lists over candidate items (see Appendix B.1 for implementation details). These models were treated purely as experimental components within our framework, and all algorithmic contributions, including the SPLiT method and the Semantic Popularity Lift (SPL) metric, were developed by the authors. In addition, LLMs (e.g., GPT-4) were used in a limited capacity to polish the writing of the paper. All scientific content, experimental design, and analysis are entirely the authors' own work, and all LLM outputs were reviewed and verified by the authors.

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

## A  SUPPLEMENTARY RELATED WORK

In this appendix, we provide additional related work to complement the discussion in Section 2. We summarize prior research on LLM-based recommender systems and on prompt optimization, both of which form the foundation of our study.

**LLM-Based Recommender Systems.** Large Language Models (LLMs) have recently been adopted as the core reasoning engines in recommender systems (Geng et al., 2022; Liu et al., 2023a; Lin et al., 2025; Zhao et al., 2024). These systems can be adapted to various recommendation tasks (Li et al., 2023), including rating prediction (Geng et al., 2022; Liu et al., 2023b), top-N recommendation (Dai et al., 2023; Hou et al., 2024; Luo et al., 2024), and sequential recommendation (Zhang et al., 2021; Liao et al., 2024). By leveraging the reasoning ability of LLMs and their access to vast amounts of background knowledge, LLM-based recommender systems can improve recommendation performance while also providing explicit, human-readable explanations for recommendations (Wu et al., 2024a; Da'u & Salim, 2020).

**Prompt Optimization.** The quality of input prompts, particularly their ability to accurately capture user preferences, plays a critical role in determining the effectiveness of LLM-based recommender systems (Mao et al., 2025; Wang et al., 2025). Early studies focused on manually designing better instructions, examples, or templates (Mirza et al., 2024; Madaan et al., 2023). However, such manual approaches are often inefficient and lack generalizability across different tasks. The advent of Automatic Prompt Optimization (APO) has alleviated these limitations by introducing automated procedures that generate optimized prompts, thus reducing manual effort while improving optimization effectiveness (Pryzant et al., 2023; Ramnath et al., 2025; Kong et al., 2025). Nevertheless, the quality of the optimized prompts is heavily dependent on the performance of the additional generation model. Inherent limitations of such models, such as popularity bias, can adversely affect the performance of the optimized prompts (Lichtenberg et al., 2024; Di Palma et al., 2025).

# B IMPLEMENTATION DETAILS

## B.1 CANDIDATE PREFERENCE SUMMARY GENERATOR

In this section, we describe how candidate prompts are generated in our system for subsequent selection. The baseline prompt template follows the design in Dai et al. (2023) and specifies the fixed input structure, including the user's interaction history and candidate items, for the recommendation LLM. The instance of this template can be seen in Appendix F.1. The LLMs used to generate these prompts are implemented by GPT-4o-mini (OpenAI et al., 2024). Building on this baseline, and inspired by both our experimental findings and the results in Kusano et al. (2024), we introduce an additional preference summary component, a structured textual segment summarizing the user's interaction history to infer potential preferences. An instance for the preference summary is in Appendix F.2. This component explicitly conveys to the recommendation LLM the genres or items the user is most likely to favor, thereby guiding it toward producing recommendations that better align with user preferences. Because the other parts of the baseline prompt template remain fixed, our optimization focuses exclusively on improving this component.

To further enhance the preference summary's ability to reflect user preferences, we adopt the self-refine approach from Madaan et al. (2023) to generate multiple semantically diverse candidate preference summaries for each user. This method iteratively refines prompts using self-feedback from the LLM, potentially improving the alignment with user preferences. We provide the implementation details of the self-refine generation strategy of our experiments in Appendix B.2. However, performance improvements are not consistent across all candidate prompts. Some variants even lead to degraded recommendation performance. This variability motivates the use of a prompt selection method to identify the most effective preference summary from the generated set.

The optimized prompt template is constructed by combining the baseline prompt template with the selected preference summary component. The detailed design of preference summary generation, experimental validation of its effectiveness, and the complete final input to the recommendation LLM are provided in the Appendix F.8.

## B.2 IMPLEMENTATION DETAILS OF SELF-REFINE GENERATION STRATEGY

We adopt the self-refine strategy of Madaan et al. (2023) to build the final preference summary. The method uses two LLM roles: a feedback LLM and a refine LLM. We proceed in three steps:

1. We first produce a candidate preference instruction from the user's interaction history using the template in Section F.3.

2. The feedback LLM critiques this instruction with the template in Section F.4; an example appears in Section F.5.

3. The refine LLM revises the instruction using the prompt in Section F.6. Applying the revised instruction to the same history yields the refined preference summary shown in Section F.7.

We can repeat Steps 2 and 3 for more rounds if needed. The refined preference summary is then inserted into the optimized prompt template in Appendix F.8.

## B.3 BM25-BASED CANDIDATE RETRIEVAL

We adopt a standard BM25 lexical retriever to construct the candidate set for each user before prompting the downstream LLM. For a given user, we hold out the most recent interaction as the test item and define the candidate pool as all items in the dataset that the user has not interacted with. Each item in the pool is represented as a short text document formed by concatenating its title and genres, followed by lowercasing, basic punctuation removal, and whitespace tokenization. To represent the user's preferences, we build a user query by concatenating the titles and genres from the user's interaction history using the same preprocessing steps. We then initialize the `BM25Okapi` retriever, following the BM25 formulation in Beaulieu et al. (1997), on the tokenized pool documents and compute a relevance score for every item in the candidate pool with respect to the user query. Items are ranked by their BM25 scores in descending order, and the top $K-1$ items are selected as candidates. Together with the held-out test item, this yields $K$ items in total, which are used by the downstream LLM for recommendation and NDCG computation. This procedure is consistent with the conventional BM25 pipeline for candidate generation and serves as an appropriate first-stage retrieval mechanism.

## B.4 REWARD ESTIMATOR DETAILS

In our experiments, we adopt a Bayesian Neural Network (BNN) as the reward estimator $\mathcal{M}$ in Section 5. The BNN is trained to predict the expected reward of a preference summary given the user context. We set the balancing parameter $\xi = 0.5$, and optimize the model using mean squared error (MSE) loss.

## B.5 ALGORITHM PSEUDOCODE

For completeness, we provide the pseudocode of our proposed SPLiT algorithm and its auxiliary components. Algorithm 1 presents the main SPLiT procedure for preference summary selection. Algorithm 2 details the computation of the Semantic Popularity Lift (SPL) penalty for a candidate preference summary, and Algorithm 3 describes how the global genre popularity counts are updated online from user contexts. These components jointly implement the SPLiT framework introduced in Section 5.

## C SPL METRIC ANALYSIS

### C.1 CONSISTENCY BETWEEN SEMANTIC POPULARITY LIFT AND POPULARITY LIFT

In this section, we provide additional evidence that in terms of showing popularity bias, the Semantic Popularity Lift (SPL) introduced in Section 4.1 is consistent with the standard Popularity Lift (PL) defined on recommendation results (Abdollahpouri et al., 2020). This consistency supports the use of SPL as an appropriate metric for capturing popularity bias at the preference summary level.

**Experimental setup.** We use 100 users from the MovieLens-1M dataset. For each user, SPL is computed from the generated preference summary, and PL is computed from the recommendation list by taking the top-3 items from the re-ranked list of 5.

---

**Algorithm 1:** Semantic Popularity Lift-based Preference Summary Selection (SPLiT)

---

**Require :** rounds $T \in \mathbb{N}_+$; candidate size $n$; candidate summary generator $G$; recommender
LLM $Q$; unknown reward function $r$; tradeoff $\lambda \geq 0$; reward estimator $\mathcal{M}$ with
PREDICT/UPDATE functions; genre set $\Gamma$; global popularity counts $\{N_\gamma\}_{\gamma \in \Gamma}$ and
total $N_{\text{tot}}$

**Initialize:** history $\mathcal{H}_0 \leftarrow \varnothing$; initialize $\mathcal{M}$; set $N_\gamma \leftarrow 0$ for all $\gamma \in \Gamma$, $N_{\text{tot}} \leftarrow 0$

1 **for** *round* $t = 1, 2, \ldots, T$ **do**
2      Observe current user context $c_t$ ;
3      Draw candidate set $\mathcal{A}_t(c_t) \leftarrow G(c_t)$ with $|\mathcal{A}_t(c_t)| = n$ ;
4      **foreach** $p \in \mathcal{A}_t(c_t)$ **do**
5          $\widehat{\mu}_t(p) \leftarrow \mathcal{M}.\text{PREDICT}(c_t, p; \mathcal{H}_{t-1})$;
6          $\xi_t(p) \leftarrow \text{COMPUTE\_SPL\_PENALTY}(c_t, p, \{N_\gamma, N_{\text{tot}}\})$ ;
7          $s_t(p) \leftarrow \widehat{\mu}_t(p) - \lambda\, \xi_t(p)$;
8      $p_t \leftarrow \arg\max_{p \in \mathcal{A}_t(c_t)} s_t(p)$;
9      Sample $q_t \leftarrow Q(p_t, c_t)$ and set $r_t \leftarrow r(q_t, c_t)$;
10     $\mathcal{H}_t \leftarrow \mathcal{H}_{t-1} \cup \{(c_t, p_t, r_t)\}$ ;            `// feedback for chosen` $p_t$ `only`
11     $\mathcal{M}.\text{UPDATE}(c_t, p_t, r_t; \mathcal{H}_t)$;
12     UPDATE\_GLOBAL\_POPULARITY\_FROM\_CONTEXT$(\{N_\gamma, N_{\text{tot}}\}, c_t)$;

---

**Algorithm 2:** Compute\_SPL\_Penalty$(c, p, \{N_\gamma\}_{\gamma \in \Gamma}, N_{\text{tot}})$

---

**Require :** context $c$; candidate preference summary $p$; semantic textual similarity model
STS : text $\rightarrow \Delta(\Gamma)$; global popularity counts $\{N_\gamma\}_{\gamma \in \Gamma}$ and total $N_{\text{tot}}$

1 $w_h(c, \gamma) \leftarrow \text{STS}(c)$ ;          `// genre distribution from context`
2 $w_{pr}(p, \gamma) \leftarrow \text{STS}(p)$ ;     `// genre distribution from preference summary`
3 **foreach** $\gamma \in \Gamma$ **do**
4      $\theta(\gamma) \leftarrow \dfrac{N_\gamma}{\max(1, N_{\text{tot}})}$
5 $E_h(c) \leftarrow \sum_{\gamma \in \Gamma} w_h(c, \gamma)\, \theta(\gamma)$;    $E_{pr}(p) \leftarrow \sum_{\gamma \in \Gamma} w_{pr}(p, \gamma)\, \theta(\gamma)$;
6 $\text{SPL} \leftarrow \frac{E_{pr}(p) - E_h(c)}{E_h(c)}$;
7 **return** SPL

---

**Algorithm 3:** Update\_Global\_Popularity\_From\_Context$(\{N_\gamma\}_{\gamma \in \Gamma}, N_{\text{tot}}, c)$

---

**Require :** genre set $\Gamma$; current counts $\{N_\gamma\}_{\gamma \in \Gamma}$ and total $N_{\text{tot}}$; context $c$ with per-genre
occurrence counts $\text{Count}_c(\gamma)$ (the number of times genre $\gamma$ appears over all
interactions in $c$)

1 $S_c \leftarrow \sum_{\gamma \in \Gamma} \text{Count}_c(\gamma)$ ;         `// total genre occurrences in c`
2 **foreach** $\gamma \in \Gamma$ **with** $\text{Count}_c(\gamma) > 0$ **do**
3      $N_\gamma \leftarrow N_\gamma + \text{Count}_c(\gamma)$ ;       `// if c contains` $\gamma$ `d times, add d`
4 $N_{\text{tot}} \leftarrow N_{\text{tot}} + S_c$ ;            `// total number of genre labels`

---

**Popularity Lift (PL).** This definition is consistent with Abdollahpouri et al. (2020), as it show the
popularity bias of recommendation results.

**Semantic Popularity Lift (SPL).** SPL is defined in Section 4.1 by comparing the global-
popularity-weighted genre distributions of the preference summary and the user history:

$$\text{SPL}(p, c) = \frac{E_{pr}(p) - E_h(c)}{E_h(c)}. \tag{5}$$

**Results.** To test consistency, we bin users into deciles by their SPL values and compute the mean
PL with standard error of the mean (SEM) in each bin. As shown in Figure 4, PL increases mono-
tonically with SPL, indicating that a higher SPL in preference summaries corresponds to more

popularity-skewed recommendation results. At the user level, we also compute correlations: SPL and PL exhibit a Pearson correlation of $r = 0.64$ and a Spearman rank correlation of $\rho = 0.43$. These significant correlations further validate the consistency between the two measures.

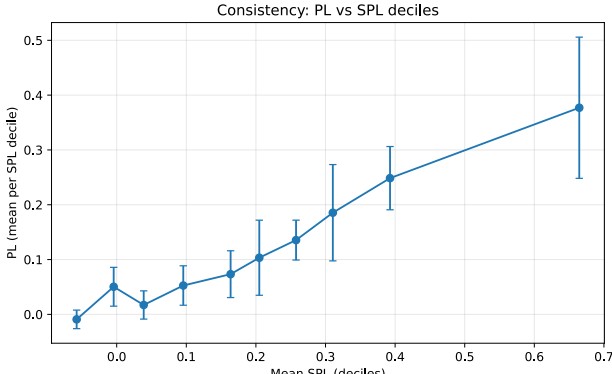

Figure 4: Consistency of SPL and PL. Users are binned into deciles of SPL, and the mean PL with SEM is reported for each bin (recommendation set: top-3 of top-5 genres). The monotonic trend demonstrates that higher SPL values in preference summaries correspond to higher PL in recommendation results.

**Conclusion.** These findings demonstrate that SPL and PL are strongly consistent: if a preference summary exhibits popularity bias as measured by SPL, the corresponding recommendation results are also very likely to display popularity bias as reflected by PL, with the strength of the bias closely aligned across the two measures. Therefore, SPL serves as a suitable and reliable metric for quantifying popularity bias in preference summaries, complementing PL measured on recommendation results.

## C.2 Analysis of Semantic Popularity Lift (SPL) on the Recommendation Performance

To further investigate the impact of popularity bias on recommendation quality, we analyze the relationship between Semantic Popularity Lift (SPL) and the ranking accuracy of the downstream LLM-based recommender. Figure 5 reports the average NDCG@3 across prompts grouped by their SPL values.

**Experimental setup.** We randomly sampled 360 users from the dataset and generated three prompts per user using the procedure described in Section 5. Each prompt was evaluated by feeding it into the downstream recommender LLM and computing the NDCG@3 of the resulting recommendation list following the evaluation protocol in Section 6. We then associated each prompt with its computed SPL value and the corresponding NDCG@3 score. To visualize the trend, all 1080 prompts were sorted by SPL and partitioned into ten quantile bins. Within each bin, we report the mean SPL on the x-axis, the mean NDCG@3 on the y-axis, and the standard error of the mean (SEM) as error bars.

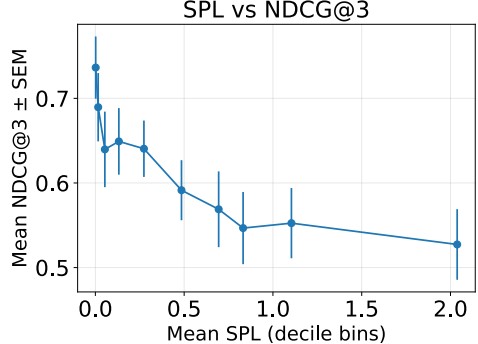

Figure 5: Relationship between Semantic Popularity Lift (SPL) and recommendation accuracy. The figure reports mean NDCG@3 across 1080 prompts (360 users $\times$ 3 prompts), grouped into ten quantile bins by SPL. Error bars indicate the standard error of the mean (SEM). Higher SPL values are associated with lower NDCG@3, revealing the negative impact of popularity bias.

**Results.**     As shown in Figure 5, we observe a clear negative correlation between SPL and recommendation performance. Prompts with the lowest SPL values (close to zero) achieve the highest NDCG@3, approximately 0.70–0.75. As SPL increases, the average NDCG@3 consistently decreases. In the mid-range (0.5–1.0), the performance drops to around 0.55–0.60, and at the highest SPL levels (near 2.0), the scores converge close to 0.53. While minor fluctuations exist in intermediate bins, the downward trajectory is monotonic. The SEM bars further indicate that the trend is statistically robust.

**Discussion.**     These results demonstrate that prompts with higher SPL, which are more semantically aligned with popular items, tend to produce worse recommendation outcomes. In contrast, prompts with low SPL preserve novelty and user-specific relevance, thereby achieving superior NDCG@3. This analysis provides empirical evidence that popularity bias directly undermines the ranking accuracy of LLM-based recommender systems and underscores the importance of SPL-aware debiasing mechanisms in prompt optimization.

# D    Experimental Setup

## D.1    Recommendation Task: Top-$K$ Recommendation

In this work, we focus on the top-$K$ recommendation task (He et al., 2023). The goal is to recommend a ranked list of $K$ items that a user is most likely to interact with, given their browsing history, profile, and other contextual information. Here, $K$ is a hyperparameter specifying the number of items recommended to each user. We adopt the widely used leave-one-out evaluation strategy in He et al. (2017) to assess recommendation performance. The detailed evaluation protocol of the leave-one-out strategy is provided in Appendix D.3.

## D.2    Dataset Description

We conduct experiments on two widely used benchmark datasets for recommender systems:

**MovieLens-1M** (Harper & Konstan, 2015). This dataset contains approximately 1 million explicit ratings from 6,000 users on 4,000 movies. Each movie is associated with metadata such as title and genres, which we use to analyze popularity bias.

**Last.fm** (Bertin-Mahieux et al., 2011). This dataset records implicit feedback in the form of music listening histories from Last.fm users. We process the user–artist interaction data to build preference histories and use artist tags (e.g., genres) as side information.

Both datasets provide rich user–item interaction histories along with genre-level metadata, enabling evaluation of recommendation performance and analysis of popularity bias.

## D.3    Leave-One-Out Evaluation

We adopt the leave-one-out strategy as our evaluation protocol for item recommendation. The motivation is to simulate the realistic scenario of predicting a user's next interaction while avoiding information leakage. By holding out only the most recent interaction as the test item, the protocol preserves the temporal order of user behavior and ensures that the model is evaluated on a genuine prediction task.

**Procedure.**     For each user, we treat the most recent interaction as the test item, and use the few preceding interactions (2-6 when available) as the user's history context. To construct a candidate set, we retrieve four negative items that the user has not interacted with, typically selected by a retrieval method such as BM25 to provide challenging distractors. These negatives are combined with the held-out test item to form a candidate set of five items. The recommender is then required to rank the entire candidate list conditioned on the user history, and evaluation is based on whether the true test item is ranked highly.

**Evaluation.** We compute Hit Ratio (HR) and Normalized Discounted Cumulative Gain (NDCG) as evaluation metrics. HR measures whether the test item appears within the top-$k$ results (with $k = 5$ in our setup), while NDCG further considers the exact rank position, rewarding higher placements.

**Benefits.** This protocol is particularly suitable for our setting because it closely mimics the real-world task of next-item recommendation and, at the same time, ensures fairness by assigning each user exactly one test case that is both comparable and consistent across the dataset.

## D.4 NDCG

**Normalized Discounted Cumulative Gain (NDCG@$k$)** In recommender system evaluation, we assume that for each user $u$ there exists a ranked list $R(u)$, which represents the ordered sequence of items produced by the recommendation algorithm. Formally, $R(u) = (i_1, i_2, \ldots, i_{|I|})$, where $i_j$ is the item at position $j$ and $|I|$ is the size of the candidate item set. Since only a limited number of items are displayed to the user in practice, we truncate this list to the top-$k$ results, denoted as $\text{Topk}(u) = (i_1, i_2, \ldots, i_k)$.

The relevance score $r_{u,i}$ indicates whether item $i$ is truly relevant for user $u$. In implicit-feedback scenarios, we typically adopt a binary definition: $r_{u,i} = 1$ if user $u$ actually interacted with item $i$ (e.g., clicked, watched, or purchased), and $r_{u,i} = 0$ otherwise.

Based on this definition, the discounted cumulative gain at cutoff $k$ for user $u$ is given by

$$\text{DCG@}k(u) = \sum_{j=1}^{k} \frac{2^{r_{u,i_j}} - 1}{\log_2(1+j)}, \tag{6}$$

where $i_j$ denotes the $j$-th item in $\text{Topk}(u)$. This formulation captures the fact that relevant items appearing at higher ranks contribute more to the recommendation quality. To characterize the ideal ranking, we define the ideal DCG at cutoff $k$ as

$$\text{IDCG@}k(u) = \sum_{j=1}^{k} \frac{2^{r_{u,i_j^\star}} - 1}{\log_2(1+j)}, \tag{7}$$

where $(i_1^\star, \ldots, i_k^\star)$ represents the ideally sorted list with all relevant items ranked before irrelevant ones. In the case of binary relevance, if user $u$ has multiple relevant items, then

$$\text{IDCG@}k(u) = \sum_{j=1}^{\min(k,|R_u|)} \frac{1}{\log_2(1+j)} \tag{8}$$

where $|R_u|$ denotes the number of relevant items for user $u$, i.e., the number of items $i$ such that $r_{u,i} = 1$ (user $u$ has actually interacted with item $i$). If no relevant items exist, IDCG is set to 1 to avoid division by zero. If no relevant items exist, we define $\text{IDCG@}k(u) = 0$ and set $\text{NDCG@}k(u) = 0$ accordingly.

The normalized metric is then defined as

$$\text{NDCG@}k(u) = \begin{cases} \text{DCG@}k(u)/\text{IDCG@}k(u), & \text{IDCG@}k(u) > 0, \\ 0, & \text{otherwise.} \end{cases} \tag{9}$$

Finally, we compute the average performance across all users:

$$\text{NDCG@}k = \frac{1}{|U|} \sum_{u \in U} \text{NDCG@}k(u). \tag{10}$$

Therefore, NDCG@k reflects the ranking quality of the top-$k$ recommended items, taking into account both the relevance of the items and their positions in the list. This makes it one of the most widely adopted evaluation metrics in recommender systems.

Table 3: Baseline algorithms used for evaluating optimization effectiveness

| Baseline | Surrogate Model | Acquisition Function |
|---|---|---|
| GP-UCB | Gaussian Process (GP) (Rasmussen, 2003) | Upper Confidence Bound (UCB) (Srinivas et al., 2009) |
| GP-EI | Gaussian Process (GP) | Expected Improvement (EI) (Jones et al., 1998) |
| BNN-UCB | Bayesian Neural Network (BNN) (Springenberg et al., 2016) | Upper Confidence Bound (UCB) |
| BNN-EI | Bayesian Neural Network (BNN) | Expected Improvement (EI) |

### D.5 HIT RATE

Hit Rate (HR) is a widely used metric to evaluate the effectiveness of a recommender system in terms of whether the relevant items appear in the top-$k$ recommended list. Formally, suppose we have a ranked list $R(u)$ generated for user $u$, and we denote the top-$k$ items in this list as $\text{Top}k(u)$. We also define the set of relevant items for user $u$ as $R_u = \{i \mid r_{u,i} = 1\}$, where $r_{u,i} = 1$ if user $u$ has interacted with item $i$, and $r_{u,i} = 0$ otherwise.

The Hit Rate for user $u$ is then defined as:

$$\text{HR@}k(u) = \begin{cases} 1 & \text{if } R_u \cap \text{Top}k(u) \neq \emptyset \\ 0 & \text{otherwise} \end{cases} \tag{11}$$

In other words, $\text{HR@}k(u)$ checks whether at least one of the items that user $u$ actually interacted with (i.e., relevant items) appears in the top-$k$ recommended list. The overall HR is computed by averaging over all users:

$$\text{HR@}k = \frac{1}{|U|} \sum_{u \in U} \text{HR@}k(u) \tag{12}$$

This metric captures the ability of the recommender system to successfully place at least one relevant item within the top-$k$ results. A higher HR indicates that users are more likely to find items they actually care about among the first few recommendations, which is especially important in practice where users usually only browse the top-ranked results.

### D.6 BASELINES OF OPTIMIZATION EFFECTIVENESS

To evaluate the effectiveness of our optimization framework, we compare it against several baseline algorithms that combine different surrogate models and acquisition functions in the Bayesian optimization paradigm. Specifically, we compare our SPLiT algorithm with four common Bayesian optimization baselines (see Table 3): (i) GP-UCB, which uses a Gaussian Process (GP) (Rasmussen, 2003) as the surrogate model with the Upper Confidence Bound (UCB) acquisition function (Srinivas et al., 2009); (ii) GP-EI, which also uses a GP but with the Expected Improvement (EI) acquisition function (Jones et al., 1998); (iii) BNN-UCB, which employs a Bayesian Neural Network (BNN) (Springenberg et al., 2016) together with UCB; and (iv) BNN-EI, which uses a BNN with the EI acquisition function.

### D.7 BASELINES OF RECOMMENDATION QUALITY

We compare our method against a set of representative baselines to evaluate the effectiveness of SPLiT in LLM-based recommender systems. These baselines include (i) standard LLM-based recommender systems that directly optimize prompts, and (ii) recently proposed approaches designed to mitigate popularity bias in LLM-based recommendation. However, it is important to note that

none of these methods explicitly address the popularity bias that arises in preference summary. This distinction highlights the novelty of SPLiT , which directly incorporates bias-awareness into the prompt selection process.

**(i)** LLM4RS (Dai et al., 2023): This work constructs prompt templates in Appendix F.1 and directly feeds them into the LLM to obtain recommendation results.

**(ii)** LLMRank (Hou et al., 2024): This work formulates recommendation as a conditional ranking task and shows that LLMs can act as zero-shot rankers. It further reveals challenges such as order sensitivity, position bias, and popularity bias, and proposes prompting strategies to alleviate them.

**(iii)** LLM4Rerank-accuracy (Gao et al., 2025): This method guides the LLM to focus on accuracy during reranking by designing prompt instructions that emphasize the semantic match between user preferences and candidate items, ensuring that the final list prioritizes relevance.

**(iv)** LLM4Rerank-fairness (Gao et al., 2025): This method introduces prompt instructions that explicitly enforce fairness during reranking, encouraging balanced exposure of different item groups and mitigating disparities across categories.

**(v)** WOK-minimization (Lichtenberg et al., 2024): While WOK-mitigate provides only limited effectiveness, WOK-minimization explores a stronger strategy that explicitly pushes the recommender towards long-tail content. To this end, WOK-minimization replaces the mitigate-instruction with: "Recommend indie, niche, or less well-known movies, avoiding mainstream blockbusters." This model is designed to study the extreme case of forcing recommendations away from popular items.

**(vi)** WOK-mitigate (Lichtenberg et al., 2024): Building upon the LLM-based recommendation setting, this work addresses the popularity bias commonly observed in LLM-generated recommendations. It introduces an additional instruction into the prompt template through prompt tuning, guiding the Recommendation LLM to mitigate popularity bias. The detailed instruction is provided in Appendix F.9.

# E  ADDITIONAL ANALYSIS AND ABLATION STUDIES

## E.1  POPULAR GENRES IN MOVIELENS-1M

Figure 6 illustrates the distribution of movie genres in the MovieLens-1M dataset. For clarity, we only show the four most frequent and four least frequent genres among the top 20 categories, with an ellipsis indicating the omitted middle ones. This figure highlights the distinction between popular and less popular genres. Prior work has shown that LLMs are more likely to memorize frequently occurring genres or items in MovieLens-1M, which makes their recommendation results more prone to popularity bias (Di Palma et al., 2025).

## E.2  VARIABILITY OF PREFERENCE SUMMARIZATION INDUCED BY LLM RANDOMNESS

To empirically validate Insight 2 (Variability induced by LLM randomness) in Section 4.2, we conduct an experiment where the same user context is provided to the summarization LLM multiple times. We generate different preference summaries due to the inherent randomness of LLM generation. While the experimental setup follows the protocol described in Section 6, here we specifically examine the variability of summarization outputs and their downstream impact.

Table 4 reports the case studies across multiple preference summaries for each representative users. We observe that:

1. The content of the generated preference summaries vary, even for the same user history.
2. The degree of popularity bias, measured by the Semantic Popularity Lift (SPL), also differs substantially.
3. Consequently, the recommendation performance (measured by NDCG@3 and HR@3) fluctuates depending on which summarization is used.

**Large-Scale Controlled Study on Causal Mechanism.**  Building on the variability analysis in Appendix E.2, where we observed substantial fluctuations across independently generated prefer-

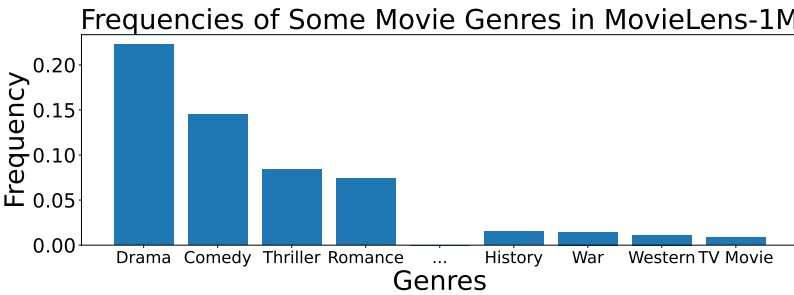

Figure 6: Frequencies of selected movie genres from the MovieLens-1M Metadata. Only the four most frequent and four least frequent genres (among the top 20) are shown, with an ellipsis indicating the omitted middle categories. This figure highlights popular genres and less popular genres in the MovieLens-1M dataset. Prior work shows that LLMs are more likely to memorize genres or items that appear more frequently in MovieLens-1M, which makes their recommendation results more prone to popularity bias (Di Palma et al., 2025).

Table 4: Variability of preference summarization across multiple LLM runs for different users. SPL and recommendation metrics (NDCG@3 / HR@3) show substantial variation across preference summaries.

| User / Preference Summary | NDCG@3 | HR@3 | SPL |
|---|---|---|---|
| User A - Preference Summary 1 | 0.00 | 0.00 | 0.45 |
| User A - Preference Summary 2 | 0.00 | 0.00 | 0.39 |
| User A - Preference Summary 3 | 1.00 | 1.00 | 0.38 |
| User B - Preference Summary 1 | 1.00 | 1.00 | 0.29 |
| User B - Preference Summary 2 | 0.63 | 1.00 | 0.35 |
| User B - Preference Summary 3 | 0.63 | 1.00 | 0.32 |
| User C - Preference Summary 1 | 0.50 | 1.00 | 0.40 |
| User C - Preference Summary 2 | 1.00 | 1.00 | 0.38 |
| User C - Preference Summary 3 | 0.50 | 1.00 | 0.43 |

ence summaries, we further conduct a large-scale controlled experiment to more rigorously isolate the causal effect of LLM stochasticity. Specifically, we fix all components of the pipeline, including user histories, candidate items, prompt templates, and evaluation protocol, and vary only the stochastic outputs of the summarization LLM. For each of the $N = 100$ users, we independently generate $M = 50$ preference summaries from the same interaction history, ensuring that any variation in popularity bias or recommendation quality arises solely from LLM randomness rather than differences in summary quality or semantic correctness.

The results in Table 5 reveal strong intrinsic instability in LLM-generated summaries. The average within-user standard deviation of NDCG@3 is $0.1831$, and $58.0\%$ of users exhibit a best–worst gap exceeding $0.5$. Popularity bias varies even more sharply: SPL shows an average standard deviation of $0.4242$, with $92.0\%$ of users having SPL ranges larger than $0.5$. These results demonstrate that even under identical conditions, LLM-generated preference summaries introduce large and systematic fluctuations in both accuracy and popularity bias.

These findings highlight that randomness in LLM outputs can lead to unstable and sometimes highly biased summarizations, which motivates the need for a selection-based strategy like SPLiT to identify higher-quality summarizations.

### E.3 CASE STUDY: POPULARITY BIAS IN PREFERENCE SUMMARIZATION

The illustration example of popularity bias in LLM-genereated preference summary is in Figure 7:

Table 5: Large-scale controlled experiment demonstrating the variability of LLM-generated preference summaries under identical conditions. All variability shown arises solely from LLM stochasticity.

| Metric | Avg. Within-User Std. | Users with Range > 0.5 |
|--------|----------------------|------------------------|
| NDCG@3 | 0.1831 | 58.0% |
| SPL | 0.4242 | 92.0% |

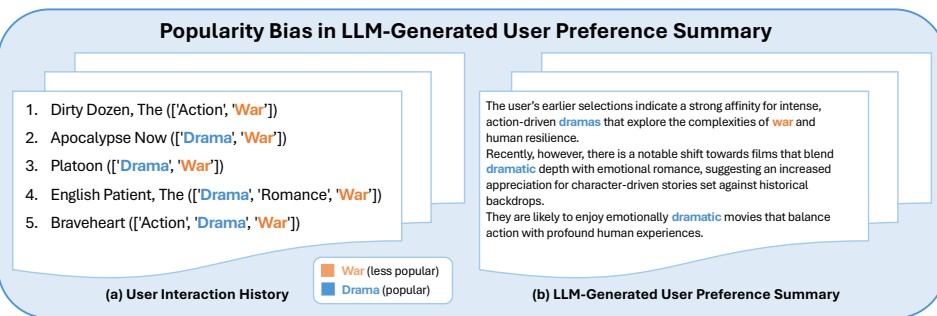

Figure 7: Illustration of popularity bias in LLM-generated user preference summary. (a) User interaction history with associated genre labels (e.g., Drama, War). (b) Preference summary generated by the LLM based on the interaction history. Although the user shows consistent interest in both Drama and War, the generated summarization tends to overemphasize Drama because it is a more popular genre in the MovieLens-1M dataset, as illustrated in Figure 6, while underrepresenting the less popular War. This imbalance highlights the tendency of LLM-based summarizations to amplify popularity bias, which can distort the downstream recommendation.

To further illustrate popularity bias in LLM-generated preference summaries, we provide a complementary example using only genre information from the interaction history, without movie titles. The input prompt follows the template in Appendix F.3, with the user history represented as:

> **Interaction History (Only Genres)**
>
> 1. (Action, War)
> 2. (Drama, War)
> 3. (Drama, War)
> 4. (Drama, Romance, War)
> 5. (Action, Drama, War)

Given this input, the LLM-generated preference summary is:

> **Generated Preference Summary**
>
> The user demonstrates a strong overall preference for **drama**-focused films, particularly those that delve into the themes of **war** and human resilience. Their earlier selections highlight an affinity for intense action-driven narratives. Recently, there has been a shift towards films that incorporate elements of **romance**, suggesting a growing appreciation for character-driven stories that maintain **dramatic** depth while exploring emotional connections against historical contexts. Overall, they favor movies that blend action with profound human experiences, with a strong emphasis on **drama** throughout their viewing history.

This example highlights how the summarization process tends to **overemphasize Drama**, a globally popular genre, while **underrepresenting War**, which appears consistently in the history but is relatively niche in the dataset. Moreover, Romance (also a global popular genre in MovieLens), only

appearing once, is amplified in the summary. This demonstrates that even when titles are removed and only genres are used as input, the summarization LLM exhibits popularity bias. Such behavior justifies the need for bias-aware preference summary selection, as formalized in our SPL metric.

### E.4 ADDITIONAL RESULTS ON REGRET CURVES

In addition to the summary results in Table 1, Figure 8 provides a detailed view of the optimization process by plotting cumulative regret over 100 decision rounds. The curves show that SPLiT consistently maintains lower regret compared with all baselines, indicating that it converges more quickly to high-quality preference summaries and makes fewer suboptimal selections throughout the process. This figure complements the table by illustrating not only the final regret values but also the convergence dynamics of different algorithms.

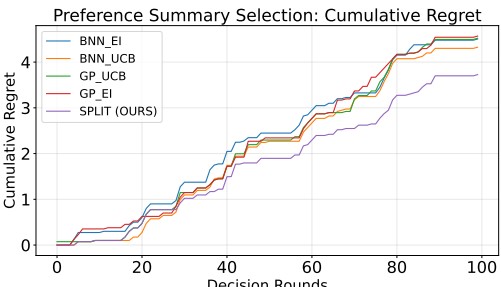

Figure 8: Optimization performance of preference summary selection algorithms. Cumulative regret of different preference summary selection algorithms over 100 decision rounds. Lower regret values indicate better optimization performance.

### E.5 ADDITIONAL EXPERIMENTAL RESULTS ON LAST.FM

To complement the main results reported in Section 6.2, we provide additional evaluation on the Last.fm dataset. As shown in Table 6, our proposed SPLiT consistently outperforms all baseline methods across NDCG@1, NDCG@3, NDCG@5, and HR@3. In particular, SPLiT improves over the strongest baseline by 7.70% in NDCG@1, 5.52% in NDCG@3, 3.13% in NDCG@5, and 1.54% in HR@3. These results demonstrate that the effectiveness of SPLiT generalizes beyond MovieLens-1M and holds across different domains.

Table 6: Performance comparison of SPLiT and baseline methods on the Last.fm dataset with GPT-4o-mini model. We report NDCG@1, NDCG@3, NDCG@5, and HR@3.

| Method | NDCG@1 | NDCG@3 | NDCG@5 | HR@3 |
|---|---|---|---|---|
| LLM4RS | 0.5152 | 0.6105 | 0.6809 | 0.6768 |
| LLMRank | 0.5979 | 0.7419 | 0.8057 | 0.8453 |
| LLM4Rerank-accuracy | 0.7273 | 0.8240 | 0.8700 | 0.8889 |
| LLM4Rerank-fairness | 0.6667 | 0.8114 | 0.8454 | 0.9192 |
| WOK-minimization | 0.3939 | 0.5552 | 0.6834 | 0.6768 |
| WOK-mitigate | 0.7273 | 0.8200 | 0.8452 | 0.8889 |
| **SPLiT (ours)** | **0.7833** | **0.8695** | **0.8972** | **0.9333** |
| *Improvement over best baseline* | *+7.70%* | *+5.52%* | *+3.13%* | *+1.54%* |

### E.6 ABLATION STUDY ON SUMMARIZATION INSTRUCTIONS

To further investigate whether modifying the instructions of the summarization LLM can improve recommendation performance, we conducted an ablation study without preference summary selection. Specifically, we replaced the default summarization instruction with several alternatives (the

details of each instruction are provided in Appendix F.12, with corresponding code snippets). Each variant was used to generate preference summarizations, which were then fed into the downstream recommender LLM under the same setup described in Section 6.

Table 7 summarizes the results in terms of NDCG@3 and HR@3. We observe that changing the summarization instruction yields only marginal differences across variants. Importantly, popularity bias persists across all cases, indicating that simply modifying instructions is insufficient to substantially improve recommendation performance or mitigate bias.

Table 7: Ablation study on different summarization instructions. The results are reported in terms of NDCG@3 and HR@3.

| Instructions | NDCG@3 | HR@3 |
|---|---|---|
| Instruction 1 | 0.5523 | 0.7200 |
| Instruction 2 | 0.5473 | 0.7100 |
| Instruction 3 | 0.5514 | 0.7200 |

### E.7 Reasons for Not Selecting Final Recommendation Results

To address the concern, we implemented a two-stage variant that adds list-level selection on top of preference-summary selection. Specifically, we first select a preference summary, then use this summary to generate multiple recommendation lists, and finally apply the same selection strategy to choose the best-performing list. On MovieLens-1M, this summary+list approach did not outperform summary-only selection (NDCG@3 = **0.6155** vs. **0.6175**). This result indicates that the selector behaves near-random, with limited learnability in this setting. As an ablation, we also evaluate selection only of the recommendation lists (without summaries), which performs substantially worse (NDCG@3 = **0.5087**) since it lacks preference summaries to improve. These findings highlight that recommendation-list selection is not as effective, and support our design choice of focusing selection at the preference-summary level.

### E.8 Additional Large-Scale Evaluation.

To further verify the scalability of our approach, we conduct an expanded evaluation on **500 randomly sampled MovieLens-1M users**, a five-fold increase from the 100-user setup in Section 6.2. As shown in Table 8, SPLiT continues to outperform all LLM-based recommender system baselines. In particular, SPLiT achieves **12.40%** improvement on NDCG@1 and **8.48%** improvement on NDCG@3 over the best baseline. The improvement magnitude closely matches the experiments with 100 users shown in Section 6.2, demonstrating that SPLiT remains stable and effective under larger evaluation scales. These results confirm the robustness and generalization ability of SPLiT.

### E.9 Additional Results on the Anime Recommendations Dataset

We additionally evaluated our method on the Anime Recommendations Database (Kaggle, Cooper Union) dataset (Cooper Union, 2016). We sampled 100 users and kept all other experimental settings identical to those in Section 6.2. The dataset is publicly available and widely used in prior works on anime recommendation.

Across all metrics, our proposed SPLiT algorithm achieves consistent improvements over all LLM-based recommender system baselines. In particular, SPLiT improves NDCG@1 by 8.60% and NDCG@3 by 6.77% over the best-performing baseline, demonstrating strong effectiveness on this new dataset and showing that SPLiT generalizes well across domains. These results lead to two observations. First, even in a different domain such as anime recommendation, LLM-based summarization still exhibits variability and optimization headroom. Second, SPLiT continues to outperform all existing LLM baselines, indicating robust cross-dataset generalization.

Table 8: Performance comparison on **500 randomly sampled MovieLens-1M users**. All experimental settings follow Section 6.2.

| Method | NDCG@1 | NDCG@3 | NDCG@5 | HR@3 |
|---|---|---|---|---|
| LLM4RS | 0.3246 | 0.5185 | 0.6225 | 0.6603 |
| LLMRank | 0.2959 | 0.5103 | 0.6183 | 0.6633 |
| LLM4Rerank-accuracy | 0.3487 | 0.5586 | 0.6571 | 0.7134 |
| LLM4Rerank-fairness | 0.3507 | 0.5563 | 0.6757 | 0.7074 |
| WOK-minization | 0.2605 | 0.4699 | 0.5805 | 0.6273 |
| WOK-mitigate | 0.3367 | 0.5650 | 0.6647 | 0.7315 |
| **SPLiT (ours)** | **0.3940** | **0.6135** | **0.7054** | **0.7750** |
| *Improvement over best baseline* | *12.40%* | *8.48%* | *4.39%* | *5.95%* |

Table 9: Results on the Anime Recommendations dataset.

| Method | NDCG@1 | NDCG@3 | NDCG@5 | HR@3 |
|---|---|---|---|---|
| LLM4RS | 0.3232 | 0.5259 | 0.6586 | 0.6768 |
| LLMRank | 0.3469 | 0.5289 | 0.6624 | 0.6735 |
| LLM4Rerank-Accuracy | 0.3535 | 0.5491 | 0.6743 | 0.6970 |
| LLM4Rerank-Fairness | 0.3333 | 0.5517 | 0.6691 | 0.7172 |
| WOK-Minimization | 0.3030 | 0.5392 | 0.6491 | 0.6869 |
| WOK-Mitigate | 0.3131 | 0.5543 | 0.6621 | 0.7071 |
| **SPLiT (ours)** | **0.3839** | **0.5918** | **0.6962** | **0.7315** |
| *Improvement over best baseline* | *8.60%* | *6.77%* | *3.25%* | *1.99%* |

# F PROMPT TEMPLATES

## F.1 BASELINE PROMPT TEMPLATE

We reproduce the baseline prompt template used in LLM4RS (Dai et al., 2023). We use it as is in all baseline runs. The prompt takes a user's viewing history and a list of candidate movies as input. The model ranks the candidates and outputs the index of the items. The full template is below.

---

**Baseline Prompt Template**

You are a movie recommender system now.
**Input:** Here is the watching history of a user: {User History}. Based on this history, please rank the following candidate movies:
(A) {Candidate Item 1} (B) {Candidate Item 2} (C) {Candidate Item 3} (D) {Candidate Item 4} (E) {Candidate Item 5} . . .
**Output:** The answer index is

---

## F.2 INSTANCE OF PREFERENCE SUMMARY

This section shows an example of the preference summary. The first code block lists the user's interaction history. The text in parentheses gives the movie genres. The second code block is the preference summary generated from that history by the generator in Appendix B.1.

---

**User Interaction History**

1. Virgin Suicides, The (['Comedy', 'Drama']).
2. Goya in Bordeaux (Goya en Bodeos) (['Drama']).
3. Snow Falling on Cedars (['Drama']).
4. Gladiator (['Action', 'Drama']).
5. Hamlet (['Drama']).

---

**Instance of Preference Summary**

The user's earlier selections indicate a strong affinity for introspective, character-driven narratives infused with emotional complexity and artistic nuance. Recently, there has been a noticeable pivot towards more structured historical dramas that delve into moral ambiguity and grand themes of honor and sacrifice. They are likely to enjoy films that blend profound emotional depth with rich period settings and dramatic stakes—while light-hearted comedies or superficial narratives may no longer resonate.

---

### F.3 OPTIMIZED INSTRUCTION PROMPT TO GENERATE PREFERENCE SUMMARY

---

**Optimized Instruction Prompt to Generate Preference Summary**

Your task is to write an optimized instruction that summarizes a user's movie preferences based on their viewing history. The history is provided in chronological order, indexed from 1 (earliest) to the most recent.
**Your output should follow this three-part structure:** 1. Summarize the user's **overall long-term preferences** based on the earlier part of the history. 2. Emphasize the user's **recent preferences**, with a focus on any noticeable shifts in tone, theme, or narrative style. 3. Predict the kind of film the user is most likely to enjoy next—considering narrative pacing, emotional depth, stylistic features, or storytelling structure.
**Guidelines:** – Use expressive and descriptive language (e.g., "philosophical period piece", "emotionally charged character study") rather than listing genres. – Highlight dominant themes or tonal patterns, especially those that have emerged recently. – You may briefly mention what types of films **should be avoided**, particularly if they contrast with recent interest. – Keep the final output under 150 tokens, structured as a single coherent paragraph.
**Example:**
*User History (indexed from 1 = earliest to last = most recent):* 1. The Virgin Suicides 2. Goya in Bordeaux 3. Snow Falling on Cedars 4. Gladiator 5. Hamlet
*Optimized Prompt Example:* "The user's earlier selections suggest a preference for introspective, emotionally rich storytelling with poetic or artistic undertones. More recent viewings reflect a shift toward structured historical narratives grounded in classical themes and moral complexity. They are now likely to enjoy period films that emphasize restrained emotional intensity, elegant visual design, and philosophical depth—while fast-paced or stylistically fragmented stories may feel misaligned."
**Now, using the following viewing history:** User History Recordings: {history}
Write an optimized instruction following the 3-part structure: overall preference → recent preference → recommendation direction. Return **only** the final instruction paragraph, with no extra explanation.

---

## F.4 FEEDBACK PROMPT TEMPLATE

---

**Feedback Prompt Template**

Your task is to evaluate the quality of a refined preference instruction generated from a user's movie viewing history. The history is provided in chronological order, indexed from 1 (earliest) to the most recent, and the current prompt attempts to summarize the user's evolving taste.

**Your evaluation should address the following aspects:** 1. Does the prompt capture the user's **overall long-term preferences** as reflected in earlier titles? 2. Does it clearly identify the user's **recent preferences**, especially any shifts in tone, theme, or narrative style? 3. Is the **recommendation direction** well aligned with recent viewing patterns? 4. Does the language use **expressive and specific descriptions** (e.g., "whimsical satire", "philosophical historical epic") instead of vague or overly broad terms? 5. Does the new prompt represent a **meaningful refinement**, not just minor rewording?

**Example:**
*User History (indexed from 1 = earliest to last = most recent):* 1. The Virgin Suicides 2. Goya in Bordeaux 3. Snow Falling on Cedars 4. Gladiator 5. Hamlet

*Refined Prompt Example:* "The user's earlier selections suggest a preference for introspective, emotionally rich storytelling with poetic or artistic undertones. More recent viewings reflect a shift toward structured historical narratives grounded in classical themes and moral complexity. They are now likely to enjoy period films that emphasize restrained emotional intensity, elegant visual design, and philosophical depth—while fast-paced or stylistically fragmented stories may feel misaligned."

**Feedback Format:** – (Positive aspect) – (Weakness or omission) – (Suggestion for improvement)

**Now evaluate the following refined prompt:** User Interaction History: {user interaction history} Optimized Prompt: {prompt}

Return **only** the bullet-point feedback under 200 tokens, with no additional commentary.

---

## F.5 INSTANCE OF FEEDBACK

---

**Instance of Feedback**

**1.** The prompt effectively captures the user's long-term enjoyment of Comedy and Drama genres, emphasizing character-driven narratives.
**2.** It highlights the shift toward youthful and romantic comedies in recent selections.
**3.** The mention of earlier films like "Confidentially Yours" and "Rocky III" does not accurately reflect their declining relevance in the user's recent viewing history.
**4.** Genre specificity could be improved; terms like "lighthearted" and "playful" are vague—more precise descriptors (e.g., "coming-of-age romantic comedies") would enhance clarity.
**5.** Suggest: Revise the prompt to focus more on recent favorites and use specific genre expressions that reflect the user's current tastes, such as "charming teen comedies" or "humorous romantic dramas."

---

## F.6 REFINE INSTRUCTION PROMPT

---

**Refine Instruction Prompt**

Your task is to refine an instruction summarizing a user's movie preferences, based on their viewing history, an earlier version of the prompt, and feedback. The history is presented in chronological order (1 = earliest to most recent).
**Your refined output should follow this three-part structure:** 1. Summarize the user's **overall long-term preferences** based on the earlier part of the history. 2. Emphasize the user's **recent preferences**, with attention to tonal or narrative shifts. 3. Predict the type of film the user is most likely to enjoy next—based on emotional tone, pacing, and storytelling structure.
**Guidelines:** – Apply the feedback directly—do **not** simply rephrase the previous prompt. – Use expressive and descriptive phrasing (e.g., "understated moral drama", "animated emotional fable") instead of naming genres. – Focus more on **recent patterns** than on overall preferences. – You may briefly note what types of films should be avoided if they conflict with recent trends. – Keep the final instruction under 150 tokens, written as a single flowing paragraph.
**Example:**
*User History (indexed from 1 = earliest to last = most recent):* 1. The Virgin Suicides 2. Goya in Bordeaux 3. Snow Falling on Cedars 4. Gladiator 5. Hamlet
*Previous Prompt:* "The user enjoys introspective and emotionally complex films that lean toward artistic and dramatic expression. Their recent interest in large-scale narratives suggests they may appreciate epic or classical themes."
*Feedback:* – Identifies early introspective taste. – Recent shift toward structured period storytelling is not clearly emphasized. – Suggest: Emphasize the user's interest in emotionally restrained but visually rich historical films.
*Refined Prompt Example:* "The user's earlier preferences point to a fondness for lyrical, emotionally nuanced storytelling with subtle visual tone. More recent titles reveal a shift toward structured, classical narratives with restrained emotional arcs and moral themes. They are now most likely to enjoy period dramas that combine quiet intensity with philosophical weight, while disjointed or fast-cut narratives may not align with their evolving taste."
**Now, using the following input:** User History Recordings: {history} Last Optimized Prompt: {initial_prompt} Evaluation Feedback: {feedback}
Refine the instruction following the 3-part structure. Return **only** the revised paragraph.

---

## F.7 INSTANCE OF REFINED PROMPTS

---

**Instance of Refined Prompts**

'The user's viewing history reveals a strong affinity for character-driven narratives within the Comedy and Drama genres, particularly favoring charming teen comedies and humorous romantic dramas. Their recent selections, such as "American Pie" and "My Tutor," showcase a clear preference for lighthearted stories that blend romance with humor, emphasizing youthful experiences and personal growth. Recommendations should focus on contemporary films that deliver engaging and witty storytelling, ideally featuring coming-of-age themes and an uplifting tone, to align with their current tastes in comedy and romance.'

---

## F.8 INSTANCE OF OPTIMIZED PROMPT TEMPLATE

This section shows an instance of the optimized prompt template. It keeps the baseline input–output format. We insert a Preference Summary into the input to summarize the user's tastes. We generate this text from the user's interaction history using the generator in Appendix B.1, and an example appears in Appendix F.2. The full template with the optimized prompt template is shown below.

> **Optimized Prompt Template**
>
> You are a movie recommender system now.
> **Input:** Here is the watching history of a user: {User History}. Based on this history, please rank the following candidate movies:
> (A) {Candidate Item 1} (B) {Candidate Item 2} (C) {Candidate Item 3} (D) {Candidate Item 4} (E) {Candidate Item 5} . . .
> {**Preference Summary**}
> **Output:** The answer index is

## F.9 ADDITIONAL INSTRUCTION OF WOK-MITIGATE

This subsection presents the additional instruction used by WOK-Mitigate (Lichtenberg et al., 2024). The instruction asks the model to match the average popularity level in the user's past viewing. We append this instruction to the baseline prompt template in Section F.1. The inputs are the user's interaction history and a list of candidate movies. The instruction and the resulting prompt template appear below.

> **Additional Instruction of WOK-Mitigate**
>
> Recommend movies that match the average popularity level of the movies the user watched in the past. For instance, if the user mostly watched blockbusters, you should recommend movies that are also blockbusters. If, on the other hand, the user watched less well-known movies, you should recommend niche movies.

> **Prompt Template of WOK-Mitigate**
>
> You are a movie recommender system now.
> **Input:** Here is the watching history of a user: {User History}. Based on this history, please rank the following candidate movies:
> (A) {Candidate Item 1} (B) {Candidate Item 2} (C) {Candidate Item 3} (D) {Candidate Item 4} (E) {Candidate Item 5} . . .
> {**Preference Summary**}
> {**Additional Instruction**}
> **Output:** The answer index is

## F.10 ADDITIONAL INSTRUCTION OF WOK-MINIMIZATION

This subsection presents the additional instruction used by WOK-Minimize (Lichtenberg et al., 2024). The instruction asks the model to actively avoid recommending highly popular or mainstream items and instead focus on indie, niche, or less well-known ones. We append this instruction to the baseline prompt template in Section F.1. The inputs are the user's interaction history and a list of candidate movies. The instruction and the resulting prompt template appear below.

> **Additional Instruction of WOK-Minimization**
>
> Recommend indie, niche, or less wellknown movies, avoiding mainstream blockbusters.

---

**Prompt Template of WOK-Minimization**

You are a movie recommender system now.
**Input:** Here is the watching history of a user: {User History}. Based on this history, please rank the following candidate movies:
(A) {Candidate Item 1} (B) {Candidate Item 2} (C) {Candidate Item 3} (D) {Candidate Item 4} (E) {Candidate Item 5} . . .
{**Preference Summary**}
{**Additional Instruction**}
**Output:** The answer index is

---

### F.11 INSTANCE OF POPULARITY BIAS IN THE PREFERENCE SUMMARY

The first box shows a user's interaction history and the associated genre labels (e.g., Drama, Romance). Based on these interactions, the LLM generates a natural-language preference summary summarizing the user's tastes (the second box). Although the user exhibits consistent interest in both Drama and Romance, the generated summarization overemphasizes the more popular genre (Drama) while underrepresenting the less popular one (Romance). This imbalance highlights the tendency of LLM-based summarizations to amplify popularity bias, which can distort the downstream recommendation process.

---

**User Interaction History**

1.Star Wars: Episode VI - Return of the Jedi (['Action', 'Adventure', 'Romance', 'Sci-Fi', 'War'])
2.Ninotchka (['Comedy', 'Romance'])
3.Algiers (['Drama', 'Romance'])
4.Man and a Woman, A (Un Homme et une Femme) (['Drama', 'Romance'])
5.Piano, The (['Drama', 'Romance'])

---

**Preference Summary Generated by LLMs**

The user's earlier choices reveal a strong inclination towards romantic narratives infused with elements of drama, often set against grand or historical backdrops.
Recently, their focus has shifted toward intimate dramas that explore complex relationships and emotional nuances.
Consequently, they are likely to appreciate films that blend rich emotional depth with stylistic sophistication, perhaps favoring poignant character studies or lyrical dramas—while escapist action-packed blockbusters may feel less appealing.

---

### F.12 SUMMARIZATION INSTRUCTIONS

In this section, we present the summarization instructions used in Appendix E.6. The Summarization Instruction I is the same as the one in Appendix F.8. The Summarization Instruction II is from Kusano et al. (2024), and the Summarization Instruction III is from Wang & Lim (2023)

**Summarization Instruction I**

Your task is to write an optimized instruction that summarizes a user's movie preferences based on their viewing history. The history is provided in chronological order, indexed from 1 (earliest) to the most recent.

**Your output should follow this three-part structure:** 1. Summarize the user's **overall long-term preferences** based on the earlier part of the history. 2. Emphasize the user's **recent preferences**, with a focus on any noticeable shifts in tone, theme, or narrative style. 3. Predict the kind of film the user is most likely to enjoy next—considering narrative pacing, emotional depth, stylistic features, or storytelling structure.

**Guidelines:** – Use expressive and descriptive language (e.g., "philosophical period piece", "emotionally charged character study") rather than listing genres. – Highlight dominant themes or tonal patterns, especially those that have emerged recently. – You may briefly mention what types of films **should be avoided**, particularly if they contrast with recent interest. – Keep the final output under 150 tokens, structured as a single coherent paragraph.

**Example:**

*User History (indexed from 1 = earliest to last = most recent):* 1. The Virgin Suicides 2. Goya in Bordeaux 3. Snow Falling on Cedars 4. Gladiator 5. Hamlet

*Optimized Prompt Example:* "The user's earlier selections suggest a preference for introspective, emotionally rich storytelling with poetic or artistic undertones. More recent viewings reflect a shift toward structured historical narratives grounded in classical themes and moral complexity. They are now likely to enjoy period films that emphasize restrained emotional intensity, elegant visual design, and philosophical depth—while fast-paced or stylistically fragmented stories may feel misaligned."

**Now, using the following viewing history:** User History Recordings: {User Interaction History}

Write an optimized instruction following the 3-part structure: overall preference → recent preference → recommendation direction. Return **only** the final instruction paragraph, with no extra explanation.

**Summarization Instruction II**

Please summarize the characteristics of this user concisely and informatively. DO NOT recommend any items, DO provide a user description. The length of the output text SHOULD be within 100 words (indexed from 1 = earliest to last = most recent): {User Interaction History}

**Summarization Instruction III**

Based on my history interaction {User Interaction History} (indexed from 1 = earliest to last = most recent), what features are most important to me when selecting movies (summarize my preferences briefly)?

