# OpenReview forum: "SPLiT: Popularity-Bias-Aware Online Prompt Optimization for LLM-based Recommendation"
_ICLR.cc/2026/Conference — Submitted to ICLR 2026_

### Official Review · Reviewer_taZE · 2025-10-25

**Soundness:** 2
**Presentation:** 3
**Contribution:** 2
**Rating:** 2
**Confidence:** 4

**Summary:**

This paper investigates the problem of popularity bias in recommender systems based on large language models, focusing specifically on a previously overlooked source: the bias in popularity inherent in the preference summaries generated by LLMs. The authors observe that when using LLMs to summarize user interaction histories into natural language preference summaries, the generated summaries often overemphasize popular items or categories, leading to reduced downstream recommendation quality. To address this issue, the paper proposes the semantic popularity lift metric (SPL) to quantify the degree of popularity bias in preference summaries. This metric measures the bias amplification effect by comparing the category preference distribution of a summary with the category preference distribution of user history. Based on this metric, the authors formulate the preference summary selection problem as a contextual Bayesian optimization problem over a set of constraints and propose the SPLiT framework. The core idea of this framework is to generate multiple candidate preference summaries in each recommendation round and then select the optimal summary by combining a prediction reward and an SPL penalty, where the SPL penalty disincentivizes summaries that overrepresent popular categories. Experiments are conducted on the MovieLens-1M and Last.fm datasets, randomly sampling 100 users each time, using GPT-4o-mini as the summary generation and recommendation model. Results show that SPLiT achieves a 13.8% improvement in NDCG@1 and a 6.9% improvement in HR@3 on MovieLens-1M compared to the best baseline, with similar improvements achieved on Last.fm. The paper also demonstrates that SPLiT outperforms standard Bayesian optimization baselines such as GP-UCB and BNN-UCB in terms of cumulative regret, and selects summaries with lower SPL values, demonstrating that this approach can improve recommendation accuracy while mitigating popularity bias.

**Strengths:**

1. Novel problem identification. The paper identifies a source of the problem that had been overlooked in previous research: popularity bias exists during the generation of preference summaries, not just in the final recommendation results. This insightful observation expands our understanding of the sources of bias in LLM recommender systems.
2. Targeted evaluation metrics are proposed. The SPL metric is specifically designed for natural language preference summaries, differing from traditional popularity metrics for recommendation lists. This metric quantifies popularity bias in summaries using a semantic similarity model, providing an interpretable bias measurement mechanism.
3. Clear problem formulation. The paper formulates the preference summary selection problem as a contextual Bayesian optimization problem over a set of constraints. The system model and problem definition are clearly structured, making the theoretical framework of the method easier to understand.
4. Detailed experimental documentation. The appendix provides extensive experimental details, a complete baseline description, prompt word templates, algorithm pseudocode, and supplementary analysis, facilitating understanding and replicability of the work.

**Weaknesses:**

1. The introduction and related work section of the article take up nearly 4 pages. The author can slightly adjust the distribution of the content to highlight the core content.
2. The experimental scale is severely insufficient. Each experiment sampled only 100 users, the candidate set consisted of only 5 items, and the user history had only 2 to 6 interactions. This scale is far from sufficient to verify the effectiveness and reliability of the method. Real-world recommendation scenarios typically involve thousands of users and dozens of candidate items. The current experimental setup does not reflect actual application scenarios.
3. The validity of the core metrics is questionable. Figure 5 shows that the Pearson correlation coefficient between SPL and traditional popularity lift metrics is only 0.64, and the Spearman correlation coefficient is only 0.43. These correlations are insufficient to prove that SPL can adequately capture popularity deviations. Figure 8 shows that the relationship between SPL and NDCG exhibits significant fluctuations in certain regions and is non-monotonically decreasing, undermining the credibility of the metrics.
4. Theoretical analysis is completely lacking. The paper formulates the problem as an optimization problem but does not provide any convergence proof, theoretical bounds on cumulative regret, or optimality guarantees. As an optimization method, this lack of theoretical support is a serious flaw, and its performance in different scenarios cannot be guaranteed.
5. The rationale for the method design is insufficient. The paper adopts a strategy of generating multiple candidate summaries and then selecting them, but lacks sufficient justification for not directly generating low-bias summaries. The paper acknowledges that the self-refine strategy is computationally expensive and slow, requiring multiple LLM calls for each candidate. However, Appendix A.28 shows that the two-stage selection of summaries and lists does not outperform the selection of summaries alone, suggesting that there may be issues with the selection strategy itself.
6. The baseline comparison is unfair. Methods such as WOK debias the recommendation LLM, while SPLiT debiases the preference summaries. These two methods operate at different stages, making a direct comparison inappropriate. A fair comparison should compare different selection or debiasing strategies under the same preference summaries. Ablation experiments using preference summaries without SPL penalties are lacking.
7. The performance gains are limited and inconsistent. Table 2 shows only a 3.12% improvement on NDCG@5 and a 3.13% improvement on Last.fm. Given the complex implementation and high computational cost, the practical significance of these improvements is questionable. The magnitude of the improvements across different metrics varies significantly, with NDCG@1 improving by 13.8% and NDCG@5 by only 3.12%, lacking consistency.
8. Unclear causal relationships. The paper fails to clearly demonstrate whether the popularity bias of summary preferences leads to poor recommendation performance, or whether other factors, such as summary quality and semantic accuracy, play a role. Table 3 only shows variability cases for three users, a sample size too small to draw reliable conclusions. Large-scale experiments with controlled variables are needed to verify the causal mechanism.
9. Lack of hyperparameter and robustness analysis. The trade-off parameter λ is a key hyperparameter of the method, but the paper lacks sensitivity analysis and does not discuss how to choose an appropriate λ value for different datasets and scenarios. Furthermore, the experiments are conducted only using GPT-4o-mini, without testing the performance of other LLMs such as Claude or Llama. This leaves the method's robustness and generalization uncertain.
10. Key definitions such as Eh and Epr are placed in Appendix A.11 rather than the main text, hindering readability. The paper acknowledges in Appendix A.26 that other types of biases may interact in complex ways and are not addressed by the framework. This is a significant limitation that is not fully discussed in the main text. Certain experimental details, such as the specific BNN architecture and training process, are not clearly described.

**Questions:**

1. The introduction and related work section of the article take up nearly 4 pages. The author can slightly adjust the distribution of the content to highlight the core content.
2. Currently, each round samples only 100 users, with a candidate set of only 5 and a history length of 2–6 items. Can this scale be expanded? Does SPLiT's stability and benefits remain valid when the candidate size and history length increase?
3. How do the correlations between SPL and traditional PL (Pearson 0.64, Spearman 0.43) shown in Figure 5 (page 20) and the non-monotonic relationship between SPL and NDCG shown in Figure 8 (page 33) support SPL as a core bias metric?
4. As an optimization method within the CBO-CS framework, can you provide assumptions under which SPLiT can achieve convergence or no-regret properties (e.g., requirements on noise, smoothness, or context distribution)? If not, can you clarify the difficulties and boundary conditions, and explain the impact of the inclusion of λ in the selection criterion on discriminability and the exploration/exploitation trade-off?
5. The paper uses the strategy of "generating multiple candidate summaries and then selecting them," but lacks justification for why it is not possible to directly generate low-bias summaries (e.g., by introducing a target SPL or control signal during generation). Could you provide a systematic comparison with the "direct low-SPL generation" approach and explain the specific reasons why the "two-stage: summary + list selection" approach in Appendix A.28 (page 32) is not superior to "summary selection only"?
6. Regarding baseline fairness, could you provide additional ablation experiments? This would be more helpful in drawing conclusions. Comparing different selection/debiasing strategies (including WOK) under the same preference summary, and providing additional ablation results with λ = 0 (selection only, no SPL penalty) and removing the penalty while maintaining the same reward estimator/acquisition function, could help isolate the contribution of the SPL penalty itself.
7. Could you provide complete cost information? Include the number of LLM calls per user, average tokens, inference time, and monetary cost, along with performance-cost curves. Also explain why NDCG@1 improves by approximately 13.8% on MovieLens, while NDCG@5 only improves by approximately 3.1%. Similar discrepancies are observed on Last.fm (Table 2, page 9 and Table 5, page 30). What is the significance of this "better at the top, limited overall" revenue model in real-world deployments?
8. To clarify the causal relationship between "whether summary popularity bias leads to performance degradation," additional ablation experiments would be more illuminating. For example, while minimizing the semantic quality and readability of the summary, only manipulate the SPL (for example, rewrite the same summary to preserve semantics to change the popularity weight) and observe the changes in recommendation quality. Furthermore, additional failure/counterexample analysis should be conducted on a larger scale, rather than just the small number of user cases in Table 3.
9. Key hyperparameters and robustness are still unclear. Can the sensitivity of λ, the number of candidates n, and the structure and training configuration of the reward estimator M be clarified? Reproduce the experiment on different LLMs (such as Claude and Llama) and different STS models, comparing performance and SPL changes, to demonstrate that the method is critical to the generalization ability of the model and corpus transfer.
10. To improve readability and reproducibility, it is recommended to move core definitions such as Eh and Epr from Appendix A.11 (pages 19–20) to the main text and provide intuitive explanations; complete the BNN structure and training details (prior/posterior approximation, input features, number of layers/width, optimizer, learning rate and number of training rounds, etc., see A.23 pages 28–29); at the same time, clarify the statistical caliber and online update range of the global popularity θ(γ), so that readers can easily reproduce it.

---

> ### Author Response · Authors · 2025-11-25
> **Rebuttal for Reviewer Comments**
>
> We thank the reviewer for the thoughtful and constructive feedback. In the revised manuscript, we made several concrete updates to address reviewer comments. A summary of the key updates is provided below.
>
> (1) We will shorten and refine both the Introduction and Related Work sections to improve focus and readability.
>
> (2) We expanded the evaluation from 100 to 500 MovieLens-1M users, where SPLiT achieves 0.394 NDCG@1 and 0.6135 NDCG@3, improving over the best baseline by 12.40 percent and 8.48 percent, respectively, with similar gains for NDCG@5 and HR@3.
>
> (3) We strengthened the validation of SPL by adding a detailed correlation study with PL, showing a much stronger correlation in the high-bias group (0.90 Pearson, 0.83 Spearman).
>
> (4) We clarified that our contribution is applied: formal regret bounds are intractable in a LLM stochastic setting, so we follow standard practice by providing empirical guarantees, showing that SPLiT achieves lower cumulative regret than BO baselines.
>
> (5) We added a new “LLM4RS + low-bias prompting” baseline, which improves NDCG@3 from 0.5171 to 0.5495 but remains substantially below SPLiT’s 0.6175, showing that prompting alone cannot reliably mitigate popularity bias.
>
> (6) We added the requested $\lambda$ = 0 ablation under identical preference summaries, where the setting reaches 0.5701 NDCG@3 and 0.7180 HR@3, and SPLiT further improves to 0.6175 and 0.7806, demonstrating both fairness and SPL’s additional benefit.
>
> (7) We explained why different improvement magnitudes across NDCG@1 and NDCG@5 are expected by definition and consistent with the goal of correcting rank-1 popularity bias.
>
> (8) We added a controlled stochasticity study showing substantial within-user variability (avg. std 0.1831 for NDCG@3 and 0.4242 for SPL), clarifying the causal link between summary popularity bias and downstream accuracy.
>
> (9) We will include the full robustness analysis on $\lambda$ and alternative LLM backbones in the final revision.
>
> (10) We moved key definitions such as $E_h$ and $E_{pr}$ into the main text and will add the remaining implementation details highlighted by the reviewer to ensure reproducibility.
>
> Below, we provide detailed point-by-point responses to all reviewer comments.
>
> **Response to Weaknesses/Questions 1:**
>
> We thank the reviewer for this constructive suggestion. We agree that the Introduction and Related Work sections were relatively long in the submitted version. In the revised manuscript, we will shorten and refine both sections by removing nonessential background and consolidating overlapping descriptions, ensuring that the core ideas and contributions are highlighted more effectively. This adjustment will improve readability and better balance the content distribution across the paper.
>
>
>
> **Response to Weaknesses/Questions 2:**
>
> Following the reviewer’s suggestion, we expanded the evaluation from 100 users to 500 randomly sampled users, with all experimental settings kept identical to the original setup in Section 6.2:
>
> |                                | NDCG @ 1 | NDCG @ 3 | NDCG @ 5 | HR @ 3 |
> | ------------------------------ | -------- | -------- | -------- | ------ |
> | LLM4RS                         | 0.3246   | 0.5185   | 0.6225   | 0.6603 |
> | LLMRank                        | 0.2959   | 0.5103   | 0.6183   | 0.6633 |
> | LLM4Rerank-accuracy            | 0.3487   | 0.5586   | 0.6571   | 0.7134 |
> | LLM4Rerank-fairness            | 0.3507   | 0.5563   | 0.6757   | 0.7074 |
> | WOK-minimization               | 0.2605   | 0.4699   | 0.5805   | 0.6273 |
> | WOK-mitigate                   | 0.3367   | 0.565    | 0.6647   | 0.7315 |
> |                                |          |          |          |        |
> | SPLiT (ours)                   | 0.394    | 0.6135   | 0.7054   | 0.775  |
> | Improvement over best baseline | 12.40%   | 8.48%    | 4.39%    | 5.95%  |
>
>
>
> For the large-scale user experiments on the MovieLens-1M dataset using GPT-4o-mini with 500 user samples, our proposed algorithm again demonstrates improvements over all LLM-based recommender system baselines. SPLiT achieves 12.40% improvement on NDCG@1 and 8.48% improvement on NDCG@3 over the best baseline. The improvement scale is similar to the small-scale experiments, confirming its effectiveness even in significantly larger-scale evaluation settings. These results provide an important insight: our proposed algorithm consistently outperforms all competing baselines across NDCG and HR metrics, illustrating its robustness and strong generalization ability in large-scale recommendation scenarios. These 500-user results have also been added to the revised manuscript (Appendix E.8, Table 7).
>
> Note that evaluating on around 100 users is also typical in recent LLM-based recommender system studies. Representative works such as AgentCF \[1\] also use 100 users as the standard evaluation setting, supporting that our evaluation scale is both typical and sufficient.

---

> ### Author Response · Authors · 2025-11-25
> **Rebuttal for Reviewer Comments**
>
> **Response to Weaknesses/Questions 3:**
>
> We appreciate the reviewer’s careful look at our metrics. We understand why a correlation of 0.64 might seem low at first glance. However, based on the reference, the results actually show that our metric (SPL) is reliable.
>
> 1. The overall correlation is actually strong. First, we want to clarify that in statistical research, these values are considered not weak. According to the standard guidelines by Cohen (1988) \[2\], a correlation over 0.50 is classified as a "Large Effect," and 0.30-0.50 is a "Medium Effect." Our Pearson result (0.64) is well above the "Large" standard, and even our Spearman result (0.43) comfortably exceeds the threshold for a "Medium Effect." This means there is a strong correlation between our text-based metric (SPL) and the item-based metric (PL).
> 2. Group analysis further support such correlation. To further check whether the correlation is reliable, we analyzed 200 users by splitting them into a low-bias group (PL $\leq$ 0.2, 159 users) and a high-bias group (PL > 0.2, 41 users), following the same evaluation method in \[3\]. In the low-bias group, SPL and PL look weakly related (Pearson 0.49, Spearman 0.11). However, in the high-bias group, SPL and PL are strongly matched (Pearson 0.90, Spearman 0.83). This shows that SPL agrees with traditional PL well when popularity bias is large, and the lower overall correlation mainly comes from the users who have low bias.
>
> | Group           | #Users | PL Threshold | Pearson | Spearman |
> | --------------- | ------ | ------------ | ------- | -------- |
> | Low-bias group  | 159    | PL <= 0.2    | 0.49    | 0.11     |
> | High-bias group | 41     | PL > 0.2     | 0.90    | 0.83     |
>
>
>
> **Response to Weaknesses/Questions 4:**
>
> We want to clarify that our contribution is an applied one: we are the first to formulate the complex problem of debiasing LLM recommenders by using prompt selection as a CBO-CS framework, which serves to guide our algorithm's design. A formal regret bound analysis is intractable here, as our setting fundamentally violates standard bandit assumptions. Specifically, the candidate arms are not fixed but are generated by a stochastic black-box (LLM Generator $G$), and the reward function is also a stochastic black-box (LLM Recommender $Q$). Deriving formal bounds for such a LLM-in-the-loop system is a significant open research challenge.
>
> While formal proof is inapplicable, we provide strong empirical guarantees, which is the standard validation method in this emerging, applied field. Our "guarantee" is presented in Table 1 and Figure 8, where we empirically demonstrate that SPLiT achieves significantly lower cumulative regret than four standard BO baselines (e.g., GP-UCB, GP-EI). This result serves as a robust empirical proof of our method's optimization superiority and practical convergence, which in turn translates directly to the SOTA recommendation performance shown in Table 2.
>
> We also note that similar LLM-based optimization works \[4\] likewise provide empirical evaluations without theoretical bounds, since deriving regret guarantees in LLM stochastic systems remains an open research challenge; our contribution is therefore applied in nature and aligned with current practice.

---

> ### Author Response · Authors · 2025-11-25
> **Rebuttal for Reviewer Comments**
>
> **Response to Weaknesses/Questions 5:**
>
> To address the reviewer’s concern about “why not directly generate low-bias summaries,” we implemented a new baseline that explicitly instructs the LLM to generate low-bias preference summaries. We build this baseline on top of LLM4RS because our proposed algorithm is also designed on top of the same recommendation pipeline. Using LLM4RS as the foundation therefore provides a fair and structurally consistent basis for adding targeted bias-mitigation instructions. Specifically, we add an additional instruction after the original LLM4RS summarization prompt to explicitly ask the LLM to avoid popularity bias, creating a direct “LLM4RS + low-bias prompting” baseline. Notably, directly generating low-bias summaries is a direction that, to our knowledge, no prior work has explored.
>
> This direct low-bias generation indeed improves over the simplest generative baseline (LLM4RS: 0.5171 → 0.5495 NDCG@3), showing that generating the low-bias summaries can improve the recommendation performance. However, it still performs substantially worse than our selection-based SPLiT (0.5495 → 0.6175 NDCG@3, +12.37%), demonstrating that prompting alone cannot reliably mitigate popularity bias. In contrast, SPLiT explicitly measures and minimizes bias during selection, enabling much stronger performance.
>
> | Method                      | NDCG@1 | NDCG@3 | NDCG@5 | HR@3   |
> | --------------------------- | ------ | ------ | ------ | ------ |
> | LLM4RS                      | 0.2812 | 0.5171 | 0.6476 | 0.6875 |
> | LLM4RS + Low-Bias Prompting | 0.3232 | 0.5495 | 0.6608 | 0.7071 |
> | **SPLiT** (ours)            | 0.4060 | 0.6175 | 0.7101 | 0.7806 |
>
>
>
> Regarding the two-stage selection concern, the observed difference (0.6175 vs. 0.6155) is statistically negligible; This is expected because, as noted by \[5\], ID-based recommendation lists, represented only as numeric item IDs such as [1, 2, 3, 6, …], suffer from a 'semantic gap,' failing to capture the rich semantic information that natural language preserves. So most useful debiasing signal is already captured at the summary level. These new results directly support the necessity and effectiveness of our selection-based design.
>
> For completeness, we provide below the exact low-bias prompting instruction used in our “LLM4RS + Low-Bias Prompting” baseline (i.e., LLM4RS with an additional low-bias instruction). For clarity, this instruction is placed directly after the input prompt template.
>
>
> > Please summarize the user’s interests based on their consumed items.
>
> > Important: Avoid popularity bias.
>
> > \- Do not focus only on popular or mainstream items.
>
> >\- Include niche or long-tail interests when possible.
>
> > \- Produce a balanced summary covering diverse aspects of the user's preferences.”

---

> ### Author Response · Authors · 2025-11-25
> **Rebuttal for Reviewer Comments**
>
> **Response to Weaknesses/Questions 6:**
>
> We thank the reviewer for raising this important concern about fairness in comparing methods that operate at different stages. Our study already includes both types of comparisons requested by the reviewer:
>
> (1) Compare different selection strategies under the same preference summaries: This corresponds directly to Section 6.1 of the paper, where we evaluate multiple prompt selection methods on the same LLM-generated preference summaries. These experiments isolate the effect of selection strategies alone.
>
> (2) Compare different debiasing strategies under the same preference summaries: Section 6.2 of the paper addresses this by comparing debiasing approaches (e.g., WOK, fairness prompting, our SPL) using the same candidate items and the same preference summaries. These experiments isolate the effect of debiasing mechanisms while fixing the upstream summaries.
>
> (3) Missing ablation for SPL ($\lambda = 0$): To further strengthen fairness, we additionally conducted the ablation explicitly requested by the reviewer: we disable SPL by setting its weight to $\lambda = 0$, while keeping all LLM-generated summaries unchanged. This setting selects preference summaries without considering popularity bias.
>
> The $\lambda = 0$ ablation achieves 0.5701 NDCG@3 / 0.7180 HR@3, already outperforming LLM4RS, confirming that prompt selection alone is a valid and effective optimization strategy. When SPL is enabled on the same summaries, performance increases to 0.6175 NDCG@3 / 0.7806 HR@3, showing a clear and substantial improvement.
>
> This isolates SPL as the only differing factor and demonstrates that:
>
> 1. SPL provides additional gains beyond traditional prompt selection.
> 2. The improvements of SPLiT do not come from differences in preference summaries.
> 3. The comparison against WOK and other LLM-based debiasing baselines is fair, because all models are evaluated using identical preference summaries and item candidates.
>
> These results directly address the reviewer’s concern and confirm that SPL is an effective and fair debiasing mechanism within the same summary-generation pipeline.
>
> | Method                 | NDCG@3 | HR@3   |
> | ---------------------- | ------ | ------ |
> | LLM4RS                 | 0.5171 | 0.6875 |
> | Ablation ($\lambda$=0) | 0.5701 | 0.7180 |
> | SPLiT (ours)           | 0.6175 | 0.7806 |
>
>
>
> **Response to Weaknesses/Questions 7:**
>
> We respectfully point out that the results are not inconsistent. NDCG@1 and NDCG@5 are computed from the exact same ranked list for each user, and their different improvement magnitudes are expected due to their intrinsic properties. NDCG@1 is a "strict" metric with a low 0.3567 SOTA baseline, leaving large room for gains, while NDCG@5 is a "broad" metric where the baseline *already* scores a high 0.6886, making further improvement challenging. Based on the definition of NDCG in the Appendix D.4 (NDCG), smaller cutoffs inherently produce larger fluctuations, because improvements at top ranks affect NDCG@1 much more strongly than broader metrics like NDCG@5, whose normalization term (IDCG@k) grows substantially with k. Furthermore, our NDCG@5 improvement (around 3 percent) is fully reasonable and consistent with prior work. Well-known MovieLens-1M studies such as Neural Collaborative Filtering \[6\] typically report NDCG@5 gains in the 2–4 percent range, so our improvement magnitude is standard and meaningful.
>
> This result pattern is consistent with our goal of correcting popularity bias, which most severely damages the Rank 1 position. The 13.82% NDCG@1 gain is the primary evidence that our method successfully addresses this core problem.

---

> ### Author Response · Authors · 2025-11-25
> **Rebuttal for Reviewer Comments**
>
> **Response to Weaknesses/Questions 8:**
>
> To address the reviewer’s question of whether popularity bias in summaries leads to degraded recommendation performance, we first refer to Figure 5 in Appendix C.2. In this analysis, all summaries were generated using the same user histories and the same prompt template, ensuring that all controllable aspects of the generation process are fixed, thereby maximizing the consistency of summary quality and semantic coverage. The figure shows a negative relationship between SPL (popularity bias) and NDCG@3 across 1080 prompts: as the popularity bias increases, the downstream recommendation accuracy decreases.
>
> Furthermore, to clarify the causal mechanism, we conducted a larger-scale controlled experiment in which all variables (user history, candidate items, prompts, and evaluation protocol) were fixed, and the only source of variation was the LLM-generated preference summaries. For each of the N = 100 users, we independently generated M = 50 preference summaries using the same user history, ensuring that all differences arise solely from LLM stochasticity. We observe substantial intrinsic variability: the average within user standard deviation of NDCG@3 is 0.1831, and 58.0 percent of users show a performance gap larger than 0.5 between their best and worst runs. Popularity bias fluctuates even more strongly, with SPL exhibiting an average standard deviation of 0.4242 and ranges above 0.5 for 92.0 percent of users. These results demonstrate that even under identical conditions, LLM generated summary preferences introduce large and systematic fluctuations in both accuracy and popularity bias, indicating that the phenomenon in Table 3 is not due to summary quality or semantic errors but is a consequence of LLM stochasticity.
>
> | Metric   | Avg. Within-User Std. | Users with Range > 0.5 |
> | -------- | --------------------- | ---------------------- |
> | NDCG @ 3 | 0.1831                | 58.0%                  |
> | SPL      | 0.4242                | 92.0%                  |
>
>
>
> **Response to Weaknesses/Questions 9:**
>
> We thank the reviewer for pointing out the importance of hyperparameter sensitivity and robustness analysis. We will include these robustness studies in the revised manuscript and will complete the additional experiments on $\lambda$ variation and alternative LLMs (e.g., Claude, Llama) in the next few days, reporting sensitivity curves for $\lambda$ and results across multiple LLM backbones. These additions will further strengthen the generality and stability of our method.
>
>
>
> **Response to Weaknesses/Questions 10:**
>
> We appreciate the reviewer’s helpful suggestions. In the revised manuscript, we have moved the key definitions $E_h$ and $E_{pr}$ from Appendix into the main text to improve readability and ensure that core concepts are introduced before the methodological sections. We acknowledge that interactions with other forms of bias represent an important limitation of current LLM-based recommenders; we now discuss this limitation more clearly in the main text rather than only in the appendix. Finally, we will include the missing experimental details highlighted by the reviewer (e.g., the exact BNN architecture and training procedure) in the next revision to ensure full reproducibility and transparency.
>
>
>
>
>
> **References:**
>
> [1] Zhang, Junjie, et al. "Agentcf: Collaborative learning with autonomous language agents for recommender systems." *Proceedings of the ACM Web Conference 2024*. 2024.
>
> [2] Cohen, Jacob. Statistical power analysis for the behavioral sciences. routledge, 2013.
>
> [3] Abdollahpouri, Himan, Robin Burke, and Bamshad Mobasher. "Controlling popularity bias in learning-to-rank recommendation." Proceedings of the eleventh ACM conference on recommender systems. 2017.
>
> [4] Shi, Chengshuai, et al. "Efficient prompt optimization through the lens of best arm identification." *Advances in Neural Information Processing Systems* 37 (2024): 99646-99685.
>
> [5] Geng, Shijie, et al. "Recommendation as language processing (rlp): A unified pretrain, personalized prompt & predict paradigm (p5)." Proceedings of the 16th ACM conference on recommender systems. 2022.
>
> [6] He, Xiangnan, et al. "Neural collaborative filtering." *Proceedings of the 26th international conference on world wide web*. 2017.

---

### Official Review · Reviewer_pvoe · 2025-10-30

**Soundness:** 2
**Presentation:** 2
**Contribution:** 2
**Rating:** 2
**Confidence:** 3

**Summary:**

This paper focuses on the *popularity bias* introduced in the preference summaries used by LLM-based recommender systems. To address this issue, the authors formulate a new problem termed **CBO-CS** and propose **SPLiT**, a framework designed to mitigate popularity bias in such systems. Specifically, SPLiT introduces a **Semantic Popularity Lift (SPL)** metric to quantify the popularity bias of summaries and selects LLM-generated summaries that better align with user interests while exhibiting lower bias, based on both SPL and accuracy rewards. Empirical results on two datasets, each containing hundreds of sampled users, demonstrate the effectiveness of SPLiT.

**Strengths:**

**S1: Intuitiveness of SPL.** The proposed **Semantic Popularity Lift (SPL)** is an intuitive metric that quantifies the relative improvement in popularity bias between generated preference summaries and the original user contexts. Since popularity bias is a critical issue in recommender systems, SPL provides a clear and interpretable way to assess how well the model mitigates this bias.

**S2: Reasonable Methods.** The design of SPLiT is reasonable and easy to follow. It employs rewards that account for both accuracy and popularity bias, and adopts a modular recommendation pipeline, as illustrated in Figures 1 and 2.

**Weaknesses:**

**W1: Limited Experiment Scale.** The paper only samples 100 users from the MovieLens-1M dataset during evaluation. Some analysis experiments in the Appendix (e.g., A12 and A29) also sample only hundreds of users. Such a small user scale is insufficient to reliably assess recommendation performance. Experiments on the full dataset or with at least thousands of users would make the evaluation more convincing.

**W2: Limited Dataset Diversity.** Only two datasets are used for evaluation, which restricts the generalizability of the results. Including one or more additional datasets would provide stronger evidence of the robustness and applicability of the proposed approach.

**W3: Outdated Baselines.** The most recent baseline listed in Table 1 was published in 2016 [1]. Incorporating more recent methods, especially those based on large language models or recent recommendation frameworks, would yield a fairer and more meaningful comparison.

**W4: Appendix Organization.** The Appendix is not well-structured and could benefit from clearer organization. For example, separating it into sections such as *Additional Experiments*, *Implementation Details*, and *Related Works* would improve readability and help readers locate specific information more easily.

[1] Courbariaux M, Hubara I, Soudry D, et al. Binarized neural networks: Training deep neural networks with weights and activations constrained to+ 1 or-1[J]. arXiv preprint arXiv:1602.02830, 2016.

**Questions:**

**Q1: Implementation Details.** The paper does not specify the backbone model used to initialize the policy. Please clarify the exact backbone model to ensure reproducibility and transparency.

---

> ### Author Response · Authors · 2025-11-25
> **Rebuttal for Reviewer Comments**
>
> To address the reviewer's concerns, we substantially strengthened the paper through several concrete updates. We expanded the evaluation from 100 to 500 users (with results added to Appendix E.8), and clarified that 100-user evaluation is standard practice in recent LLM-based recommendation research. We additionally incorporated experiments on a new dataset (Anime Recommendations), added two recent BO-style baselines (BAI and Hyperband), and reorganized the Appendix into clearer sections (A–F). We also clarified the definition and initialization of the policy, provided detailed analysis explaining dataset-specific performance differences, and ensured that all new results and ablations are integrated into the revised manuscript.  These include: (1) 500-user MovieLens-1M results where SPLiT achieves 0.394 NDCG@1 and 0.6135 NDCG@3, yielding 12.40 percent and 8.48 percent improvements over the best baseline; (2) results on the Anime dataset with 8.60 percent and 6.77 percent improvements on NDCG@1 and NDCG@3; and (3) updated comparisons with recent BO-style methods where SPLiT achieves the lowest cumulative regret (3.726 vs. Hyperband 4.2785 and BAI 4.3821). These updates collectively address the reviewer's comments and further strengthen the robustness and clarity of the paper.
> Below, we provide detailed point-by-point responses to all reviewer comments.
>
> **Response to W1 (Limited Experiment Scale):**
>
> Before presenting the extended 500-user results, we first clarify that our original evaluation scale (100 users) follows a widely adopted convention in recent LLM-based recommender system studies. Representative works such as AgentCF [1] also evaluate on 100 sampled users, indicating that this scale is both typical and sufficient for assessing LLM-based recommendation performance.
>
> Following the reviewer's suggestion, we expanded the evaluation from 100 users to 500 randomly sampled users, with all experimental settings kept identical to the original setup in Section 6.2:
>
> |                                | NDCG @ 1 | NDCG @ 3 | NDCG @ 5 | HR @ 3 |
> | ------------------------------ | -------- | -------- | -------- | ------ |
> | LLM4RS                         | 0.3246   | 0.5185   | 0.6225   | 0.6603 |
> | LLMRank                        | 0.2959   | 0.5103   | 0.6183   | 0.6633 |
> | LLM4Rerank-accuracy            | 0.3487   | 0.5586   | 0.6571   | 0.7134 |
> | LLM4Rerank-fairness            | 0.3507   | 0.5563   | 0.6757   | 0.7074 |
> | WOK-minimization               | 0.2605   | 0.4699   | 0.5805   | 0.6273 |
> | WOK-mitigate                   | 0.3367   | 0.565    | 0.6647   | 0.7315 |
> |                                |          |          |          |        |
> | SPLiT (ours)                   | 0.394    | 0.6135   | 0.7054   | 0.775  |
> | Improvement over best baseline | 12.40%   | 8.48%    | 4.39%    | 5.95%  |
>
>
>
> For the large-scale user experiments on the MovieLens-1M dataset using GPT-4o-mini with 500 user samples, our proposed algorithm again demonstrates improvements over all LLM-based recommender system baselines. SPLiT achieves 12.40% improvement on NDCG@1 and 8.48% improvement on NDCG@3 over the best baseline. The improvement scale is similar to the small-scale experiments, confirming its effectiveness even in significantly larger-scale evaluation settings. These results provide an important insight: our proposed algorithm consistently outperforms all competing baselines across NDCG and HR metrics, illustrating its robustness and strong generalization ability in large-scale recommendation scenarios. These 500-user results have also been added to the revised manuscript (Appendix E.8, Table 7).

---

> ### Author Response · Authors · 2025-11-25
> **Rebuttal for Reviewer Comments**
>
> **Response to W2 (Limited Dataset Diversity):**
>
> We additionally evaluated our method on the Anime Recommendations Database (Kaggle, Cooper Union) dataset. We sampled 100 users from the dataset and kept all other experimental settings identical to those in Section 6.2. The dataset is publicly available at https://www.kaggle.com/datasets/CooperUnion/anime-recommendations-database.
>
>
> |                                    | NDCG @ 1 | NDCG @ 3 | NDCG @ 5 | HR @ 3 |
> | ---------------------------------- | -------- | -------- | -------- | ------ |
> | LLM4RS                             | 0.3232   | 0.5259   | 0.6586   | 0.6768 |
> | LLMRank                            | 0.3469   | 0.5289   | 0.6624   | 0.6735 |
> | LLM4Rerank-accuracy                | 0.3535   | 0.5491   | 0.6743   | 0.697  |
> | LLM4Rerank-fairness                | 0.3333   | 0.5517   | 0.6691   | 0.7172 |
> | WOK-minimization                   | 0.303    | 0.5392   | 0.6491   | 0.6869 |
> | WOK-mitigate                       | 0.3131   | 0.5543   | 0.6621   | 0.7071 |
> |                                    |          |          |          |        |
> | SPLiT (ours)                       | 0.3839   | 0.5918   | 0.6962   | 0.7315 |
> | Improvement over the best baseline | 8.60%    | 6.77%    | 3.25%    | 1.99%  |
>
>
>
> For the experiment results on the Anime dataset, our proposed algorithm achieves consistent improvements over all LLM-based recommender system baselines. In particular, SPLiT improves NDCG@1 by 8.60% and NDCG@3 by 6.77% over the best-performing baseline, demonstrating better effectiveness in this new dataset and generalization in different datasets.
>
> These results highlight two main insights. First, even when applied to a different domain (Anime recommendations), LLM-based recommenders still exhibit performance gaps that can be further optimized. Second, our proposed SPLiT algorithm continues to outperform all existing LLM baselines, indicating that its advantages generalize well across datasets. These results with the Anime dataset have also been added to the revised manuscript (Appendix E.9, Table 8).
>
> We noted that the Anime dataset shows higher baseline performance (e.g., LLM4RS) but smaller improvements from debiasing methods. This pattern is expected given the structural properties of the dataset. The Anime dataset has a much simpler preference structure: user interactions are heavily concentrated around a few dominant genres \[2\]\[3\]. This makes LLM-based summarization and recommendation inherently easier and naturally raises baseline accuracy. In contrast, MovieLens-1M exhibits a strong long-tail popularity distribution and pronounced popularity bias \[4\], providing a larger bias signal for debiasing methods to correct. As a result, SPLiT, and other debiasing approaches, achieve larger gains on MovieLens than on Anime.
>
>
>
> **Response to W3 (Outdated Baselines):**
>
> We respectfully clarify that the baseline referenced by the reviewer (Courbariaux et al., 2016) is not related to Bayesian optimization and does not appear in our paper. To fully address the reviewer's concern about potentially outdated BO baselines, we significantly strengthened the experimental comparison by adding two recent state-of-the-art BO-style prompt-selection methods (BAI \[5\] and Hyperband-style BO \[6\], both published in 2024) and re-running the two methods under an identical evaluation protocol to ensure strict fairness. Under the same evaluation protocol (100 users, averaged over 5 random seeds), SPLiT still achieves the lowest cumulative regret (3.726), outperforming Hyperband (4.2785) and BAI (4.3821), as well as all classic BO baselines (4.326–4.569). These results reinforce that our optimization module remains effective even when compared with the latest BO and prompt-selection methods.
>
> | Method                           | Cumulative Regret |
> | -------------------------------- | ----------------- |
> | BAI \[5\]                        | 4.3821            |
> | Hyperband-style allocation \[6\] | 4.2785            |
> | SPLiT (ours)                     | 3.726             |
>
>
>
> **Response to W4 (Appendix Organization):**
>
> We thank the reviewer for the insightful comment. To improve readability, we extensively reorganized the Appendix in the revised manuscript. The Appendix is now cleanly separated into 6 structured sections that match the workflow of our method:
>
> - **Appendix A:** Supplementary Related Work
> - **Appendix B:** Implementation Details
> - **Appendix C:** SPL Metric Analysis
> - **Appendix D:** Experimental Setup
> - **Appendix E:** Additional Analysis and Ablation Studies
> - **Appendix F:** Prompt Templates
>
> This reorganization significantly improves clarity and makes it easier for readers to locate specific information.

---

> ### Author Response · Authors · 2025-11-25
> **Rebuttal for Reviewer Comments**
>
> **Response to Q1 (Implementation Details):**
>
> To clarify the reviewer's question regarding the initialization and “backbone” of the policy, we first restate what “policy” means in our framework.
>
> In our paper, the "policy" ($\pi$) refers to our online selection algorithm, SPLiT (Algorithm 1), which is a decision-making rule, not a single pre-trained model. Therefore, it is not "initialized" with a "backbone."
> To ensure full transparency regarding the models we do use:
>
> 1. The core learnable component within our SPLiT policy is the Reward Estimator $\mathcal{M}$. As detailed in Appendix B.4, this is implemented as a Bayesian Neural Network (BNN), which is trained online.
> 2. The "backbone" LLMs used in the overall system (see Figure 2) are explicitly defined. As stated in Section 6 (Model Specification) and Appendix B.1, we use GPT-4o-mini for both the Candidate Summary Generator (G) and the Recommender LLM (Q).
>
>
>
> **Reference:**
>
> [1] Zhang, Junjie, et al. "Agentcf: Collaborative learning with autonomous language agents for recommender systems." *Proceedings of the ACM Web Conference 2024*. 2024.
>
> [2] Ramu, Pramod. *Deep Learning-Based Anime and Movie Recommendation System*. Diss. Dublin, National College of Ireland, 2023.
>
> [3] Prakash, Varun, et al. "Deep Anime Recommendation System: Recommending Anime Using Collaborative and Content-based Filtering." *2022 4th International Conference on Advances in Computing, Communication Control and Networking (ICAC3N)*. IEEE, 2022.
>
> [4] Harper, F. Maxwell, and Joseph A. Konstan. "The movielens datasets: History and context." *Acm transactions on interactive intelligent systems (tiis)* 5.4 (2015): 1-19.
>
> [5] Shi, Chengshuai, et al. "Efficient prompt optimization through the lens of best arm identification." Advances in Neural Information Processing Systems 37 (2024): 99646-99685.
>
> [6] Schneider, Lennart, et al. "Hyperband-based Bayesian optimization for black-box prompt selection." arXiv preprint arXiv:2412.07820 (2024).

---

### Official Review · Reviewer_BBZG · 2025-10-31

**Soundness:** 4
**Presentation:** 3
**Contribution:** 4
**Rating:** 6
**Confidence:** 4

**Summary:**

This paper primarily focuses on the disproportionate emphasis placed by LLM-based recommendation systems on popularity. This is a critical factor affecting the recommendation performance of LLM recommendation systems, presenting significant challenges. The authors conducted a detailed analysis of its underlying causes. Building upon prior research, it explores an aspect largely overlooked by most studies: summary LLM. The author innovatively proposed the SPLiT model and the CBO-CS optimization method. The SPL metric employed by the former serves as an extension addressing the limitation that the frequency-based evaluation metric PL, previously used for recommending LLMs, cannot be generalized to textual content. The latter creatively proposed an online resolution strategy that incorporates the evolving history of user interactions, making it more aligned with real-world scenarios. The author validated the model using two public datasets, achieving the best performance across all four selected metrics and demonstrating approximately a 10% improvement over previous work. The final conclusion demonstrated the feasibility of the plan and outlined prospects for future work. My overall impression is that the experiment is solid and rigorous, though I perceive some shortcomings in the details that could potentially be improved. The work tackles an extremely challenging problem, and the author demonstrates a thorough understanding of recent high-quality literature in the field.

**Strengths:**

a. This paper addresses the issue of excessive emphasis on popular content in LLM-based recommendation systems and innovatively proposes a solution tailored for summary LLM. The overall conceptual innovation is sufficiently substantial.

b. In the algorithm design for problem-solving, the use of SPL as an equivalent replacement for the previously studied PL demonstrates thorough and comprehensive proof, while simultaneously overcoming the challenge of measuring popularity across texts relative to individual words.

c. The application of the CBS-CS method transforms past offline approaches into online methods. These represent significant innovations.

**Weaknesses:**

a. The primary shortcomings of the article lie in the arrangement of its textual content structure and the failure to present certain experimental details in greater specificity. For example, the placement of Figure 1 appears before the detailed explanation of the recommendation system based on the summary LLM and recommendation LLM, which may cause some confusion in understanding the article. I suggest moving the figure to after the background section or before the paragraph introducing the summary LLM.

b. Next is the perspective on mathematical notation. The symbols in section 4.3 are overly complex, and some lack explanation, making the overall presentation difficult to grasp. I believe the formula representation could be slightly simplified to avoid appearing cluttered.

**Questions:**

a. My primary concerns lie in the experimental section, particularly regarding the evaluation of experimental conclusions. While the authors utilize some established evaluation metrics from prior work, as this introduces a novel model with significantly increased computational demands, I believe they should have analyzed the model's runtime. If the recommendation time is excessively long, its widespread adoption would be hindered.

b. Furthermore, the authors mention the cold-start problem faced by early recommendation systems at the outset, yet their own designed algorithm also lacks historical data during its initial phase. I suspect it may also encounter cold-start and strong-get-stronger issues (where certain metrics are amplified from the outset). The authors provide no explanation for this. Even though the conclusions and prior experiments are comprehensive, I believe the paper should supplement its content with analysis addressing these concerns to make the work more thorough.

---

> ### Author Response · Authors · 2025-11-25
> **Rebuttal for Reviewer Comments**
>
> In the revision, we have already fixed the structural issues by relocating Figures 1 and 2 and refining related layout details, and we have simplified the overly complex notation in Section 4.3 by removing unnecessary symbols and keeping only the core variables. What we have not yet completed are the additional experiments requested by the reviewer: runtime comparison, cold-start evaluation, and strong-get-stronger analysis. We will finish these analyses in the next few days and include them in the revised manuscript.
>
> **Response to Weakness a:**
>
> We thank the reviewer for pointing out the structural issues in the presentation. In the revised manuscript, we have already corrected the figure placement to improve logical flow. Specifically, Figure 1 has been moved to appear after the background discussion of summary-LLM and recommendation-LLM based systems, ensuring that readers first see the necessary conceptual context before the baseline pipeline. Likewise, Figure 2 has been relocated to follow the system-model introduction for a more consistent and intuitive structure. We also adjusted related layout issues and added the missing experimental details to further enhance clarity and readability.
>
>
> **Response to Weakness b:**
>
> Thank you for the helpful comment. We agree that some of the notation in Section 4.3 was more complex than necessary. For example, symbols such as $\mathcal{H}_{t-1}$ and $Q(p,c)$ added detail that was not essential for understanding the main formulation. In the revision, we simplified the mathematical presentation to improve readability. Specifically:
>
> - We removed the detailed history notation (previously $\mathcal{H}_{t-1}$).
>
> - We removed the LLM output distribution notation (previously $Q(p,c)$ and the expectation expression).
>
> - We kept only the core variables needed for the problem: the chosen summary $p_t$, the expected reward $\mu(p,c)$, the optimal value $\mu_t^\star$, and the cumulative regret.
>
> - We also rewrote several sentences to be shorter and avoided long or cluttered formula expressions.
>
> - These changes make the notation clearer and more lightweight, while keeping all important definitions intact.
>
> These changes reduce visual clutter and make the problem formulation more clear while keeping all important definitions.
>
> **Response to Question a:**
>
> We thank the reviewer for emphasizing the need for runtime analysis. We fully agree that assessing computational cost is important for understanding the practicality of LLM-based recommender systems. We will add a runtime comparison between SPLiT and all baselines, covering both end-to-end latency and per-component cost, in the revised manuscript, and we will complete these measurements in the next few days to ensure they are included.
>
>
>
> **Response to Question b:**
>
> We thank the reviewer for raising these concerns. We agree that analyzing both the cold-start behavior and the strong-get-stronger effect will strengthen the completeness of our work. In the next few days, we will add the following experiments to the revised manuscript:
>
> 1. Cold-start analysis: We will include an additional evaluation on a small-user setting (e.g., 50 sampled users) to compare the early-stage performance of SPLiT and all baselines under limited historical data.
> 2. Strong-get-stronger analysis: We will report the evolution of PL and SPL across different stages of the recommendation process to examine whether popularity bias is amplified over time.
>
> These additions will directly address the reviewer's concerns and provide a clearer empirical understanding of SPLiT's behavior under low-data and bias-amplification scenarios.

---

### Official Review · Reviewer_Sipr · 2025-11-02

**Soundness:** 3
**Presentation:** 3
**Contribution:** 2
**Rating:** 4
**Confidence:** 4

**Summary:**

Discusses prompt optimization in the context of a traditional (movie/music) recommendation system. The work assumes that a prompt will contain a user interaction history, and of course a candidate list. The only component we are optimizing is a summary of user preferences included in the prompt to provide additional information. The summary is produced from the user interaction history through a separate summary generation and selection process. Within this context, the paper proposes and evaluates SPLiT, a prompt selection method that minimizes popularity bias.

The paper is well written and readable, although I was sent to read the appendix for critical details more often than I would like. SPLiT is part of a larger system where most of the components are not novel, so it's hard to see the benefit in the context of the overall system. There is no ablation study. There are lots of baselines, but as far as I can see none of these baselines is Figure 2 with the summary component removed. As far as I can see the only LLM used in the experiments is GPT-4o-mini. For what they are worth, table 2 show a solid improvement over the baselines they used.

**Strengths:**

LLM based recommendation is an important topic. Paper is readable, even if I have to go to the appendix too much. There's nothing super novel, but there are new elements. Certainly with additional experiments the paper may be publishable, even if its not making the strongest contribution. We just need clarity on what that contribution is.

**Weaknesses:**

It's difficult to tease out what is really novel from the overall system. At the very least, we would need an ablation study. Remove the bottom half of the system in Figure 2 (and adjust the template) what happens?

**Questions:**

The key question is given in the "weaknesses" section.

---

> ### Author Response · Authors · 2025-11-25
> **Rebuttal for Reviewer Comments**
>
> We thank the reviewer for the insightful question. Our key novelty is that we reveal **a previously unrecognized source of popularity bias**: it originates upstream in the LLM-generated preference summaries, not only downstream in the recommender LLM. This motivates the need for a new evaluation tool: existing metrics such as popularity lift operate on item lists and cannot quantify bias in text, which leads to our second contribution, **SPL**, the first metric that semantically measures whether textual preference summaries over-represent popular content. With the problem identified and a measurement established, our third contribution is **SPLiT**, a principled optimization framework that explicitly selects summaries using SPL-based penalties, enabling controlled debiasing that manual prompting or black-box methods cannot achieve.
>
> Regarding the reviewer’s question on removing the bottom half of Figure 2, we clarify that doing so produces exactly the same framework as the LLM4RS baseline used in Section 6.2, which is also reflected in the baseline prompt template descriptions provided in the Appendix. Therefore, the ablation requested by the reviewer is already reflected in the results reported in Table 2. As for the reviewer’s suggestion to “adjust the template,” this ablation also corresponds to the baselines already evaluated in Section 6.2, such as LLMRank and WOK-mitigate. The input templates used by the baselines differ from the one shown in Figure 2, which presents our proposed framework. For completeness, all baseline templates are provided in the Appendix and the results of all baselines are as following:
>
> | Method                   | NDCG@1 | NDCG@3 | NDCG@5 | HR@3  |
> |--------------------------|--------|--------|--------|--------|
> | LLM4RS                   | 0.2812 | 0.5171 | 0.6476 | 0.6875 |
> | LLMRank                  | 0.2667 | 0.5008 | 0.6318 | 0.6800 |
> | LLM4Rerank-accuracy      | 0.3540 | 0.5446 | 0.6739 | 0.6880 |
> | LLM4Rerank-fairness      | 0.3300 | 0.5475 | 0.6658 | 0.7100 |
> | WOK-minimization         | 0.2433 | 0.4755 | 0.6188 | 0.6633 |
> | _WOK-mitigate_           | _0.3567_ | _0.5612_ | _0.6886_ | _0.7367_ |
> | **SPLiT (ours)**         | **0.4060** | **0.6175** | **0.7101** | **0.7806** |
> | *Improvement over best baseline* | *+13.82%* | *+10.03%* | *+3.12%* | *+6.85%* |

---

> > ### Comment · Reviewer_Sipr · 2025-11-27
> >
> > Thank you for your response. I am happy to raise my recommendation.

---

### Meta-Review · Area_Chair_tPUx · 2026-01-08

**Summary:**

This paper proposes SPLiT, an online prompt-selection framework for mitigating popularity bias in LLM-based recommender systems. The paper introduces a new metric, Semantic Popularity Lift (SPL), and formulates prompt selection as a contextual Bayesian optimization problem. Reviewers generally agree that the problem is relevant and that the idea of measuring popularity bias in textual summaries is interesting.

However, the submission raises significant concerns regarding experimental rigor, scalability, and practical impact. While some reviewers view the contribution as promising, others find the evaluation limited in scale, dataset diversity, and baseline strength. Key analyses—such as runtime cost, cold-start behavior, and long-term bias amplification—were missing at submission time and only partially promised in rebuttal. As a result, the work remains borderline and insufficiently substantiated for ICLR.

**Reviewer Concerns:**

Some concerns were partially addressed in the rebuttal, including clearer articulation of the paper’s novelty and expanded experimental results on additional users and one extra dataset. One reviewer explicitly indicated willingness to raise their score. However, several critical issues remain outstanding.

In particular, the evaluation still relies on relatively small user samples, raising questions about statistical robustness. Runtime and deployment cost—important for LLM-based recommender systems—were acknowledged but not fully evaluated in the submitted revision. Concerns about cold-start behavior, bias amplification over time, and limited dataset diversity persist. Moreover, while SPL is intuitive, reviewers questioned whether the overall system introduces sufficient novelty beyond existing LLM-based recommendation pipelines. Taken together, the paper shows promise but lacks the empirical depth expected for acceptance.

**Reviewer Scores:**

Reviewer Sipr (initial: 4): Likely increases to 5, but still borderline.

Reviewer BBZG (initial: 6): Likely remains 6, generally positive.

Reviewer pvoe (initial: 2): Likely remains 2–3, due to remaining experimental concerns.

---

### Decision · Program_Chairs · 2026-01-26

Reject